# Ground motion prediction maps using seismic microzonation data and machine learning

Federico Mori[1], Amerigo Mendicelli[1], Gaetano Falcone[1], Gianluca Acunzo[1], Rose Line Spacagna[1], Giuseppe Naso[2], Massimiliano Moscatelli[1]

[1] CNR-IGAG, Consiglio Nazionale delle Ricerche, Istituto di Geologia Ambientale e Geoingegneria, Area della Ricerca di Roma 1, Via Salaria km 29.300, 00015 Monterotondo (Roma), Italy

[2] Presidenza del Consiglio dei Ministri, Dipartimento della Protezione Civile (DPC), via Vitorchiano 2, 00189 Roma, Italy

*Correspondence to*: Federico Mori (federico.mori@igag.cnr.it)

**Abstract.** Past seismic events worldwide demonstrated that damage and death toll depend on both the strong ground motion (i.e., source effects) and the local site effects. The variability of earthquake ground motion distribution is caused by local stratigraphic and/or topographic setting and buried morphologies (e.g., irregular sub-interface between soft and stiff soils), that can give rise to amplification and resonances with respect to the ground motion expected at the reference site. Therefore, local site conditions can affect an area with damage related to the full collapse or loss in functionality of facilities, roads, pipelines, and other lifelines. To this concern, the *near real time* prediction of ground motion variation over large areas is a crucial issue to support the rescue and operational interventions. A machine learning approach was adopted to produce ground motion prediction maps considering both stratigraphic and morphological conditions. A set of about 16,000 accelometric data and about 46,000 geological and geophysical data were retrieved from Italian and European databases. The intensity measures of interest were estimated based on 9 input proxies. The adopted machine learning regression model (i.e., Gaussian Process Regression) allows to improve both the precision and the accuracy in the estimation of the intensity measures with respect to the available *near real time* predictions methods (i.e., Ground Motion Prediction Equation and shaking maps). In addition, maps with a 50 m x 50 m resolution were generated providing a ground motion variability in agreement with the results of advanced numerical simulations based on detailed sub-soil models.

## 1 Introduction

Spatial distributions of ground motion induced by seismic events should be properly estimated to support risk mitigation policies over large areas. Moreover, seismic risk analysis, extended to spatially distributed anthropic systems, presents new challenges in characterising the seismic risk input, regarding the spatial correlation of the ground motion values where the spatial correlation is the spatial characteristics of the ground motion arising from similarities in the seismic wave paths and local-site effects. The ShakeMaps (Wald et al., 2021), provided by the US Geological Survey, is used globally for post-earthquake emergency management and response, engineering analyses, financial instruments, and other decision-making activities. Moreover, in Italy post-event ShakeMaps are delivered by the National Institute of Geophysics and Volcanology

(Michelini et al., 2019; ShakeMap, 2021). Such ShakeMaps are based on Ground Motion Prediction Equation (GMPE; Bindi et al., 2011, among the others) and data recorded from accelerometric stations when available.

Recently, artificial intelligence-based procedures were proposed to produce *near real time* ground motion in terms of acceleration time histories (Jozinović et al., 2021; Tamhidi et al., 2021) and Intensity Measure (briefly, IM; Kubo et al., 2020, among the others). In general, ground motion maps were generated using earthquake source parameters (location, magnitude, and the finite fault if available), IM (Peak Ground Acceleration, Peak Ground Velocity, and Spectral acceleration, briefly named PGA, PGV, and Sa, respectively) at the recording accelerometric stations and the mean shear wave velocity in the upper 30 m, $V_{S30}$, as a proxy to account for site lithostratigraphic amplifications. Having shaking maps only when the first location and magnitude estimation are available, Jozinović et al. (2021) propose to use waveforms to predict the ground motion intensity by means of a Machine Learning (briefly, ML) approach (i.e., it utilizes only a training set of earthquake waveforms recorded at a pre-configured network of recording stations). Moreover, ML has been adopted to produce seismic amplification factors maps, as in the Japan case study proposed by Kim et al. (2020), rather than to provide ground motion maps. Finally, Zhou et al. (2020) propose a seismic topographic effect prediction model.

Overall, the above-mentioned works have pointed out what follows:

- hypocentral depth (H), epicentral distance (R), and magnitude (M) are widely used to estimate ground motion over large areas considering the source effect; moreover, H, R, and M are provided few minutes after an earthquake;

- $V_{S30}$, the fundamental frequency of the deposit ($f_0$), and the depth to the engineering bedrock ($H_{800}$) are the key-parameters which well gauge the effect of local sub-soil conditions on the seismic wave propagation (i.e., lithostratigraphic effect). The only $V_{S30}$ was used in the adopted ML approach since the Italian $V_{S30}$ map was provided by Mori et al. 2020a while national $f_0$ and $H_{800}$ maps are not currently available;

- elevation (h), topographic gradients ($h_x$ and $h_y$, where x and y are two orthogonal directions), and second-order topographic gradients ($h_{xx}$ and $h_{yy}$) are proxies which allow to describe the morphological effects on the seismic amplification phenomena. In this view, this work focuses on the improvement of ground motion prediction over large areas by using ML technique. The main task of this work is to suggest a procedure including all the main key-parameters together (i.e., H, R, M, $V_{S30}$, h, $h_x$, $h_y$, $h_{xx}$, $h_{yy}$).

Damage pattern induced by seismic events is related to both geological/geomorphological conditions and vulnerability of structures and infrastructures (Brando et al., 2020; Fayjaloun et al., 2021; Mori et al., 2020b, 2019). The ground motion prediction (i.e., seismic site response) is generally evaluated by means of numerical simulations which are time consuming and require well detailed models capable of properly representing sub-soil and topographic conditions (see for example, Bouckovalas and Papadimitriou, 2005; Falcone et al., 2020a, 2020b, 2018; Gatmiri and Arson, 2008; Gazetas, 1982; Luo et al., 2020; Moscatelli et al., 2020b; Pagliaroli et al., 2014; Pitilakis et al., 1999; Régnier et al., 2016, 2018).

Hence, ML approach was adopted to:

*i*) implement H, R, and M parameters available few minutes after a seismic event;

*ii*) include both lithostratigraphic ($V_{S30}$) and morphological effects (h, $h_x$, $h_y$, $h_{xx}$, $h_{yy}$);

*iii*) capture the spatial correlation at short distances (hundreds of meters) due to local site effects, which is essential for
66        reliable hazard assessments.

The main results of these elaborations are ground motion prediction maps (i.e., PGA, PGV, Sa) with the resolution of
50 m x 50 m, which can reproduce the variability captured by advanced numerical modelling.
Seismological data (i.e., H, R, M, PGA, PGV, and Sa) retrieved from European and Italian networks (Luzi et al., 2016, 2019,
2020), geological, geophysical, and geotechnical data from seismic microzonation (hereafter SM) studies (DPC, 2021), and
morphological data (ALOS, 2021) are presented in § 2. The ML approach is discussed in § 3. In detail, the § 3.1 is focused on
the adopted ML approach in terms of training and validation phase. Performances, presented in terms of Root Mean Square
Error (RMSE) and residuals (i.e., difference between the base-10 logarithms of observed and predicted values of PGA, PGV,
and Sa), are compared to the results proposed by other studies (Jozinović et al., 2021; Michelini et al., 2019, Bindi et al., 2011).
For the seismic sequence that hit Central Italy in 2016-2017, ML results and maps are shown in § 3.2 and § 4, respectively.
Referring to the seismic event occurred in Central Italy on October 30, 2016, a test is proposed in § 3.2 in terms of residuals
of the ground motion IMs (i.e., PGA, PGV, and Sa). Ground motion prediction maps for the Central Italy event occurred on
August 24, 2016, (i.e., the first destructive event of the Central Italy seismic sequence for which a great amount of studies have
been published) are shown in § 4 to enlighten the capability of the proposed ML approach to gauge the ground motion
variability at the urban scale. Moreover, with reference to § 4, the ground motion profiles, based on the proposed ML approach,
are compared with results obtained by means of two completely different methodologies: 2D numerical modelling of seismic
site response (Gaudiosi et al., 2021; Giallini et al., 2020; Grelle et al., 2020) and with the mean values predicted by the Italian
ShakeMap (2021).

## 2 Input and output data for machine learning training and validation

The input and output data for the training of ML approach, were classified into three categories: seismological, geophysical,
and morphological data. The ML approach was based on 15,779 seismological data regarding the log-10 geometric mean of
the horizontal component (geoH) for each IM (i.e., PGA, PGV, and Sa at 0.3 s, 1.0 s, and 3.0 s). Each value recorded by the
accelerometric station, named output data in Table 1 (i.e., data to be reproduced by means of ML), represents an observed
datum. In addition, Table 1 lists the used 9 predictors, named input data. Fig. 1 shows the location of the selected accelerometric
stations. Figs. 2 and 3 show the input and output data, respectively, adopted for the training phase of the selected ML approach
and presented in this section. Furthermore, some data distributions seem to be imbalanced (e.g., magnitude, M, and elevation,
h, Fig. 2). An imbalanced training input dataset is characterised by an unequal distribution of values. For instance, focusing on
Fig. 2 and M distribution, it results that the first and third quartile are 4.1 and 5.1, respectively. Moreover, focusing on elevation
distribution, it results that the first and third quartile are 136 m and 761 m, respectively. Consequently, when the ML algorithm
learns the imbalanced data (see for example, Kubo et al., 2020) the learning focus is mainly on the fit of ground motions with
magnitude lower than 6 or on the fit of site characterised by elevation lower than 1,200 m. The imbalance of the selected

| 97 | training input dataset seems to be caused by a sampling bias since no high magnitude ground motions were registered by the |
| 98 | available accelerometric stations and since few accelerometric stations have been installed at high elevation where the |
| 99 | exposition at seismic event is very low. Hence, the training dataset cannot actually be improved. In addition, distributions of |
| 100 | topographic gradients and $V_{S30}$ are characterised by few data with respect to steep slopes and high $V_{S30}$ values. How to handle |
| 101 | the imbalanced dataset in the regression problem was out of the scope of this work. Consequently, referring to a range of an |
| 102 | input datum, it is expected that the lower the number of training data the higher the uncertainty. To this end, referring to the |
| 103 | output data, maps of standard deviation are reported in § 4. |

**Table 1. Input and output data for ML training and validation.**

| Type of data | Category | Control Factors | | Database | Ref. |
|---|---|---|---|---|---|
| **INPUT** | Seismological | H | hypocentral depth | Seismological DB | Luzi et al., 2016 and 2020 |
| | | M | moment magnitude | | |
| | | R | epicentral distance | | |
| | Geophysical | $V_{S30}$ | the time-averaged shear-wave velocity to 30 m depth | Seismological DB or $V_{S30}$ map | Luzi et al., 2016 and 2020 DPC, 2021 Mori et al., 2020a |
| | Morphological | h | elevation | ALOS World 3D-30m DEM | ALOS, 2021 |
| | | $h_x$ | first order partial derivative dx (E-W slope) | | |
| | | $h_y$ | first order partial derivative dy (N-S slope) | | |
| | | $h_{xx}$ | second order partial derivative dxx | | |
| | | $h_{yy}$ | second order partial derivative dyy | | |
| **OUTPUT** | Seismological | PGA | Peak Ground Acceleration | Seismological DB | Luzi et al., 2016 and 2020 |
| | | PGV | Peak Ground Velocity | | |
| | | $Sa_{0.3}$ | Spectral acceleration at 0.3 s | | |
| | | $Sa_{1.0}$ | Spectral acceleration at 1 s | | |
| | | $Sa_{3.0}$ | Spectral acceleration at 3 s | | |

*Seismological parameters*
Seismological parameters are retrieved from Italian and European databases. Regarding 1,435 recording accelerometric
stations, PGA, PGV, spectral accelerations (i.e., Sa at 0.3 s, 1 s, and 3 s), H, R, and M were retrieved from European Strong
Motion Database, briefly ESM, (Luzi et al., 2016; ESM, 2021) and ITalian ACcelerometric Archive, herein ITACA, (Luzi et
al., 2019). In detail, data regarding the Central Italy earthquake occurred on the 2016 and recorded by temporary network
named 3A have been archived only in the ITACA database (ITACA, 2021). It is worth noting that Greek and Turkish seismic
events data were collected to consider earthquake characterised by M value greater than 6.5 and up to 7.6. Moreover,
earthquake characterised by H, R, and $\log_{10}$PGA value greater than 30 km, 400 km, and 2 ($cm/s^2$), respectively, were selected.
It should be noted that the ITACA and ESM selected data consider the shallow active crustal region (i.e., SACR zone
characterised by shallow events, H < 35 km, in agreement to Michelini et al., 2019). The distributions of seismological data of
the chosen events are shown in Figs. 2 and 3. The same figures also show the distribution of data described in the next part of
this section.

*Geophysical data*
Dynamic site condition was described by means of the time-averaged shear-wave velocity ($V_S$) to a depth of 30 meters, the
$V_{S30}$ parameter. It is worth noting that the $V_{S30}$ parameter has been successfully adopted to gauge lithostratigraphic effect on
seismic wave propagation by Falcone et al. (2021). $V_{S30}$ data (i.e., input data in ML approach), determined by means of *in situ*
investigations, are also archived in the ESM and ITACA databases. $V_{S30}$ values were retrieved from Mori et al. (2020a) for
ESM and ITACA sites not characterised by *in situ* surveys. Fig. 2 shows the distribution of the $V_{S30}$ data.
The $V_{S30}$ map proposed by Mori et al. (2020a), based on SM studies, was adopted here. The SM studies have been carried out
for the Italian municipalities through the funds allocated after the 2009 L'Aquila earthquake, in the framework of the Italian
program for seismic risk prevention and mitigation (Moscatelli et al., 2020a). Approximately 4,000 SM studies have been
already planned, representing about 99.8% of the municipalities eligible for funding (i.e., having 475 years return period
PGA ≥ 0.125g). Out of the 4,000 planned SM studies, about 75% have been completed and approved (DPC, 2021). The SM
studies permitted to collect, classify, and archive geological, geophysical, and geotechnical data with a uniform approach
following national standard criteria (SM Working Group, 2008; TCSM, 2018). The data from *in situ* tests are organised into a
database and georeferenced through an appropriate geographic information system (DPC, 2021). About 35,000 borehole logs
and 11,300 $V_S$ profiles, related to about 1,700 Down-Hole and 9,600 MASW tests, were extracted from the SM dataset. Starting
from the 11,300 $V_S$ profiles, $V_{S30}$ values were calculated. Mori et al. (2020b) derive a large-scale $V_{S30}$ map for Italy, starting
from the global morphological classes after Iwahashi et al. (2018), by integrating the large amount of data from the Italian SM
dataset. The $V_{S30}$ map by Mori et al. (2020a) was used here to integrate data where site-specific information was not available.



*Morphological data*
The morphological elevation h (i.e., an input morphological datum) was retrieved by the Advanced Land Observing Satellite
(ALOS) World 3D-30m (herein AW3D30) digital elevation model (DEM). The free version of the DEM (ALOS, 2021)
adopted here has 1 arcsec resolution, which is equivalent to approximately 30 m at the Equator. AW3D-30m global DEM data
were produced using the data acquired by the Panchromatic Remote Sensing Instrument for Stereo Mapping operated on the
ALOS from 2006 to 2011. The Japan Aerospace Exploration Agency, that is the operator of the satellite, produced the global
DEM using approximately 3 million images. Considering that AW3D30 model is the digital surface model which represents
the canopy top and building roofs' elevations, Caglar et al. (2018) found that AW3D30 is the most accurate DEM among other
similar data elevation products freely available. In detail, it was shown that the AW3D30 root mean square error is equal to
1.78 m.
Finally, a GRASS GIS command *r.slope.aspect* (https://grass.osgeo.org) was used to generate the other morphological proxies
(i.e., $h_x$, $h_y$, $h_{xx}$, and $h_{yy}$). Such command generates raster maps of first and second order partial derivatives from a raster map
of true elevation values (i.e., AW3D30 data in this study). Fig. 2 shows the distribution of the selected morphological data.

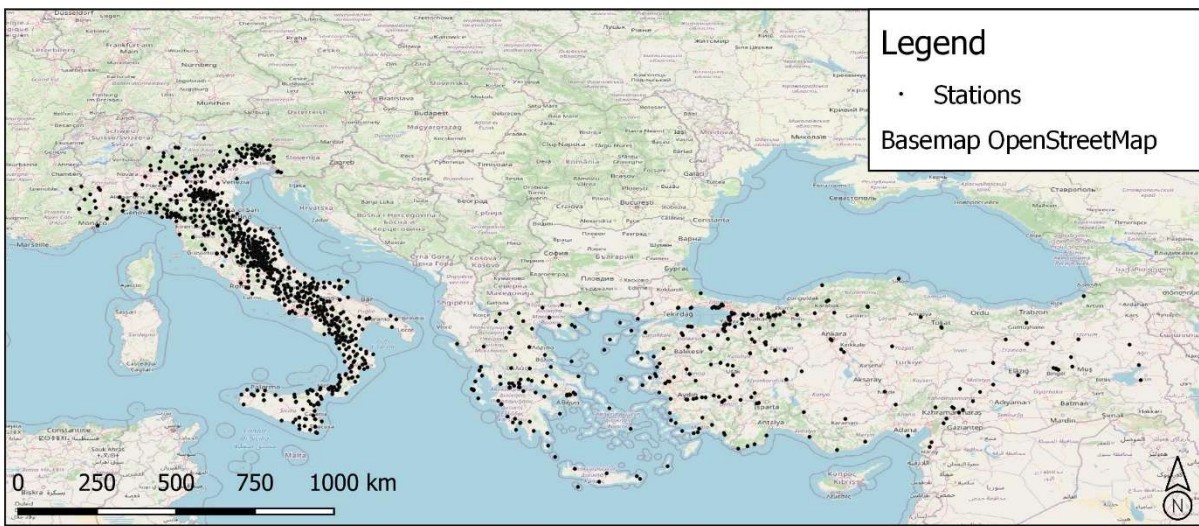


**Figure 1. Location of selected dataset (i.e., 1,435 accelerometric stations). © OpenStreetMap Distributed under the Open Data Commons Open Database License (ODbL) v1.0.**


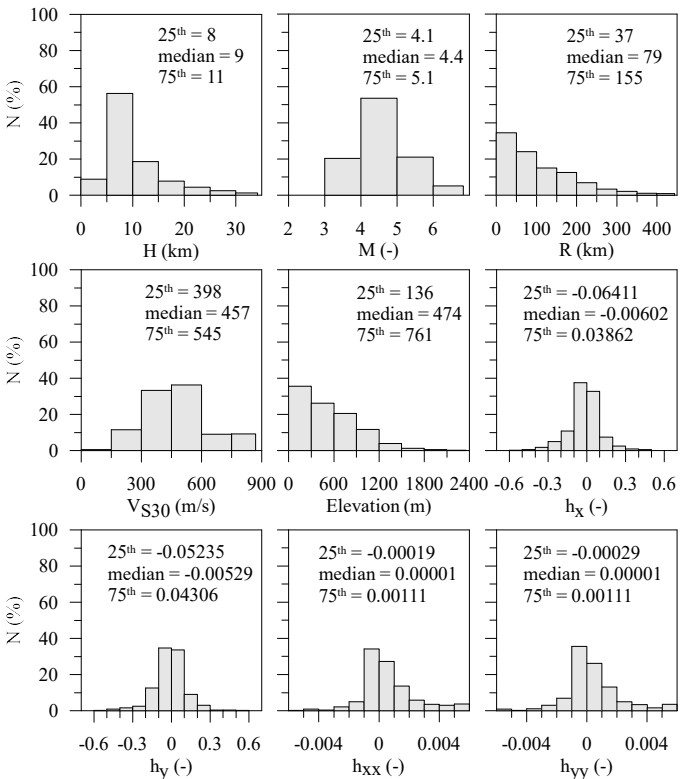


**Figure 2. Distribution of input data for the training dataset.**

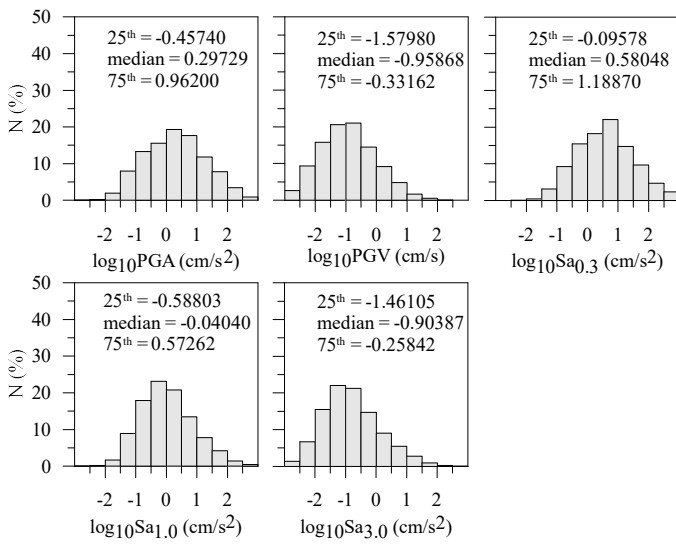


**Figure 3. Distribution of output data for the training dataset in terms of geoH IMs.**

## 3 Method

The "Matlab Regression Learner App" tool (https://it.mathworks.com/help/stats/regression-learner-app.html) was employed to produce ground motion prediction maps using a supervised ML approach. With this application, users can choose the desired models among many different methods to automatically train and validate regression models. After training multiple models, they can be compared to choose the best one. The application includes commonly used regression methods such as linear regression models, decision trees, support vector machines, ensembles of tree models, and Gaussian Process Regression (GPR). Fig. 4 shows the adopted ML workflow. After having imported and selected the data (input variables and output variables), the training and validation phases begin. In these phases the ML model that will be used is "adapted" or rather the algorithm is adapted to the training dataset. One of the objectives of this phase is the tuning of the model, acting on the hyperparameters (parameters whose value is used to control the learning process) of the algorithm to minimize errors. The K-fold cross-validation technique was used in this work. The models included in "Matlab Regression Learner App" tool have all been tested. The fitting performance (in term of RMSE) on the validation set was considered as an indicator for the generalization ability of models. Among the available models the best fitting performance in terms of RMSE was provided by the GPR model with exponential kernel (Table 2). GPR is a nonparametric, Bayesian approach to regression, which provides uncertainty measurements on the predictions. Moreover, a detailed description of GPR method is outside the scope of this work. Suggested references for comprehensive descriptions of the GPR method are Rasmussen and Williams (2006) and chapter 6 of MathWorks (2019). The above-mentioned k-fold cross-validation (k = 5) method is described in chapter 24 of Mathworks (2019).

The second step is to test the model with the best performance (GPR with exponential kernel in this research) adopting a dataset not included in the training and validation phases. The dataset for the 30 October 2016 seismic event was used since the accelerometric data of many accelerometric stations are available. The test is used to evaluate the accuracy of the model in terms of residuals (Eq. 1). In the workflow of Fig. 4 there is also a phase (comparison) that is not part of the standard ML methodology. The comparison with the ground shaking obtained by completely different methodologies was used to further analyse the ML model in terms of ground motion resolution and variability.

Training and cross-validation phases are described in § 3.1. Comparison in terms of residuals with the performance of the existing methods (i.e., an external test) is presented in § 3.2. The comparison with the ground shaking obtained by completely different methodologies is presented in § 4.

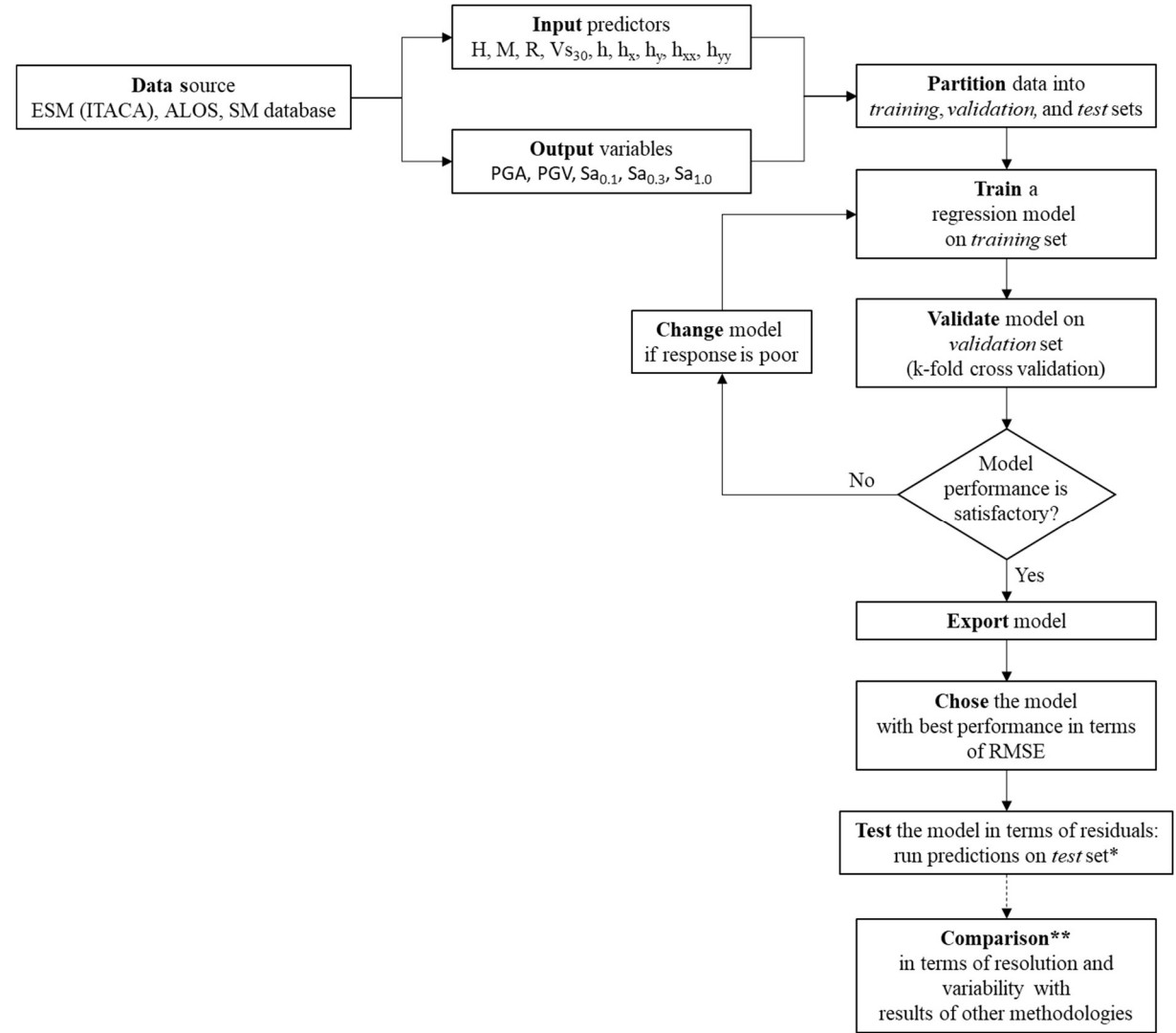

**Figure 4. ML workflow adopted in this study.**
***The selected test set was the input and output data for the October 30, 2016 seismic event.**
**\*\* This element of the workflow is not part of the standard ML methodology. This element was introduced to enlighten the**
**capability of the adopted ML procedure in estimating local scale ground motion variability. Comparison against predictions from**
**ShakeMap and 2D numerical simulations was based on August 24, 2016 seismic event input and output data.**

**3.1 Training and validation phases**
The mean RMSE of the five cross-validation datasets were adopted to select the best ML approach. With reference to the tested
ML approaches, Table 2 lists the RMSE values for each predicted IM.

**Table 2. RMSE, for all ML prediction models used to forecast log10 geometric horizontal mean (geoH) of PGA, PGV, and Sa at**
**0.3 s, 1.0 s, and 3.0 s. Suggested reference for comprehensive descriptions of the ML prediction models is MathWorks (2019).**

| ML Prediction Model | Performance in term of RMSE | | | | |
|---|---|---|---|---|---|
| | PGA | PGV | Sa(0.3s) | Sa(1.0s) | Sa(3.0s) |
| Linear Regression (Linear) | 0.53 | 0.47 | 0.50 | 0.44 | 0.43 |
| Linear Regression (Interactions Linear) | 0.48 | 0.43 | 0.47 | 0.42 | 0.40 |
| Linear Regression (Robust Linear) | 0.53 | 0.47 | 0.50 | 0.44 | 0.43 |
| Stepwise Linear Regression (Stepwise Linear) | 0.48 | 0.43 | 0.47 | 0.42 | 0.40 |
| Tree (Fine Tree) | 0.42 | 0.38 | 0.42 | 0.39 | 0.38 |
| Tree (Medium Tree) | 0.40 | 0.36 | 0.40 | 0.38 | 0.36 |
| Tree (Coarse Tree) | 0.40 | 0.36 | 0.40 | 0.37 | 0.36 |
| Support Vector Machine (Linear) | 0.53 | 0.48 | 0.49 | 0.44 | 0.43 |
| Support Vector Machine (Quadratic) | 0.43 | 0.39 | 0.42 | 0.39 | 0.39 |
| Support Vector Machine (Cubic) | 0.40 | 0.36 | 0.40 | 0.37 | 0.36 |
| Support Vector Machine (Fine Gaussian) | 0.48 | 0.46 | 0.48 | 0.45 | 0.46 |
| Support Vector Machine (Medium Gaussian) | 0.37 | 0.34 | 0.38 | 0.35 | 0.34 |
| Support Vector Machine (Coarse Gaussian) | 0.43 | 0.39 | 0.42 | 0.39 | 0.38 |
| Ensemble (Boosted Trees) | 0.40 | 0.36 | 0.40 | 0.37 | 0.36 |
| Ensemble (Bagged Trees) | 0.33 | 0.31 | 0.33 | 0.31 | 0.31 |
| Gaussian Process Regression (Squared Exponential) | 0.38 | 0.35 | 0.39 | 0.36 | 0.35 |
| Gaussian Process Regression (Matern 5/2) | 0.37 | 0.34 | 0.38 | 0.34 | 0.34 |
| **Gaussian Process Regression (Exponential)** | **0.31** | **0.30** | **0.33** | **0.30** | **0.29** |


Referring to the best prediction model (i.e., GPR with exponential kernel) and to the training dataset, Fig. 5 shows the
comparison between predicted and observed values.

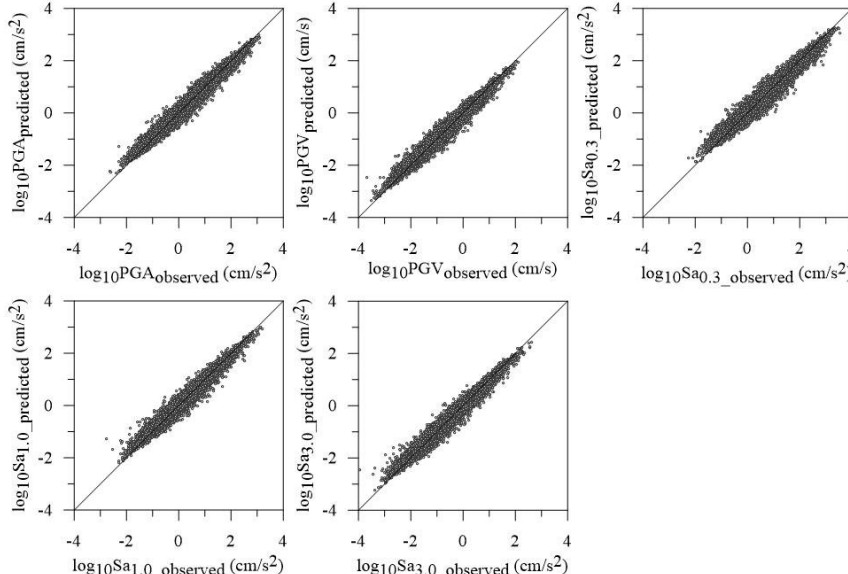


**Figure 5. Comparison between observed and predicted values referring to the output data (i.e., geoH in terms of PGA, PGV, Sa$_{0.3}$,**
**Sa$_{1.0}$, and Sa$_{3.0}$).**
The performance of the GPR model is also presented in terms of mean value and standard deviation of the residuals'
distributions (Table 3), where the residual is defined according to the Eq. (1) in agreement to what presented by other
researchers (Bindi et al., 2011; Jozinović et al., 2021; Michelini et al., 2019). It should be noted that mean and standard
deviation of the residuals' distributions referred to ShakeMap and GMPE were retrieved from the work of Jozinović et al.
(2021) to evaluate the performance of the ML approach suggested in this study. It is worth noting that the suggested ML
approach provides the best performance with respect to the approaches proposed by the other studies in terms of both accuracy
(mean value) and precision (standard deviation). In detail, the standard deviation values are reduced by the 45-60%.

$$\text{residual} = \log_{10}\left(\frac{IM_{observed}}{IM_{predicted}}\right) \tag{1}$$

**Table 3. Referring to the training dataset (15,779 data for each IM), comparison of mean and standard deviation values of the**
**residuals' distributions obtained in this study and that reported by other works (geoH stays for geometric mean of the horizontal**
**components).**

| IM (geoH) | This study (ML) | | ShakeMap | | GMPE | |
|---|---|---|---|---|---|---|
| | mean | std | mean | std | mean | std |
| PGA | -0.000033 | 0.161 | 0.038 | 0.372 | 0.017 | 0.352 |
| PGV | -0.000015 | 0.156 | 0.041 | 0.380 | -0.151 | 0.330 |
| $Sa_{0.3}$ | 0.000024 | 0.192 | 0.046 | 0.370 | -0.252 | 0.359 |
| $Sa_{1.0}$ | 0.000028 | 0.160 | 0.017 | 0.374 | -0.198 | 0.303 |
| $Sa_{3.0}$ | -0.000072 | 0.159 | -0.012 | 0.404 | 0.083 | 0.368 |


Fig. A1 in the Appendix A1 shows the contribution of each predictor variable to the reduction of standard deviation of the
residuals' distribution.
**3.2 Testing phase**
Input and output data for the October 30, 2016 seismic events were selected as external test dataset not included in the training
data. The seismic events in Central Italy of 2016 and 2017, began in August 2016 with epicentres located between Latium,
Marche, and Umbria Regions. The first strong shock occurred on August 24, 2016, at 3:36 a.m. and had a magnitude of 6.0,
with its epicentre located along the Tronto River valley, between the small municipalities of Accumoli and Arquata del Tronto.
Two powerful replicas took place on October 26, 2016, with epicentres on the Umbria-Marche border, the first shock with
magnitude 5.4 and the second with magnitude 5.9. On October 30, 2016, the strongest quake was recorded, with a moment
magnitude of 6.5 with its epicentre in Umbria Region. On January 18, 2017, a new sequence of four strong tremors with a
magnitude greater than 5 (with a maximum of 5.5) and epicentres located in Abruzzi Region took place. This set of events
caused a total of about 41,000 displaced persons, 388 injured, and 303 deaths.
In detail, the paper refers to the October 30, 2016 mainshocks since according to the available data much more accelerometric
data are available and it is therefore possible to make more detailed and reliable analyses.
Mean and std values of the residuals' distributions are presented in this section for the seismic event occurred on October 30,
2016 (briefly named test event), because it is the event with the most recordings of the whole dataset (241 accelerometric
stations). It is worth noting that this event was not included in the dataset adopted for the training phase of the ML approach.
Noting that 943 seismic events were characterised by $M \leq 6$ and 25 earthquakes by $M > 6$ (see Fig. 3 for the training dataset),
the Central Italy earthquake occurred on October 30, 2016 ($M = 6.5$) provides a robust test of the adopted ML approach. The
GMPE proposed by Bindi et al. (2014) (hereafter also Bindi GMPE) was selected to estimate the IMs at the 241 sites of interest
aiming to compare the GMPE and this ML approach performances. It should be noted that the Bindi GMPE provides IMs
depending on the $V_{S30}$ as in this study. Furthermore, the OPENQUAKE software (Pagani et al., 2014) was used to determine
the IMs values based on the selected GMPE.
Mean and std values regarding the test event (Table 4), are higher than those referred to the training and validation phase
(Table 3), as expected, because the GPR model is trained on few events with high magnitudes as discussed in § 2.
Moreover, mean and std values obtained in this example are lower than those obtained by means of GMPE as shown in Table
4. In detail, the standard deviation values are reduced by the 20-30%. Therefore, the overall performance of the proposed ML
approach is satisfactory also at the highest magnitude.

**Table 4. Comparison of mean and standard deviation values of the residuals' distributions obtained in this study and by means of**
**GMPE (Bindi et al., 2014), regarding the earthquake occurred on October 30, 2016, (241 data for each IM; geoH stays for**
**geometric mean of the horizontal components).**

| IM (geoH) | This study | | GMPE | |
| --- | --- | --- | --- | --- |
| | mean | std | mean | std |
| PGA | 0.0019 | 0.30 | -0.19 | 0.43 |
| PGV | 0.0130 | 0.34 | -0.16 | 0.42 |
| $Sa_{0.3}$ | 0.0170 | 0.32 | -0.18 | 0.39 |
| $Sa_{1.0}$ | -0.0550 | 0.35 | -0.38 | 0.46 |
| $Sa_{3.0}$ | -0.0360 | 0.39 | -0.23 | 0.55 |


**4 Ground motion prediction map for August 24, 2016 seismic event of Central Italy and comparison with numerical modelling**

After having demonstrated the goodness of the proposed method to reproduce IM values, this chapter presents examples of predictive maps produced by means of the exponential GPR model with a 50 m x 50 m resolution. In this section the map for the August 24, 2016 seismic event of Central Italy is produced to compare some significant IM profiles produced with independent advanced numerical simulations and data retrieved from ShakeMaps (2021).

The ground motion prediction map of the $Sa_{0.3}$ reported in Fig. 6 is one of the cartographic results of this study; maps of PGA, PGV, and other spectral ordinates are in the supplementary materials. Macroseismic intensities, I_MCS, retrieved by Galli et al. (2017) are also reported next to the name of the villages in Fig. 6. These maps were chosen because the 0.3 s period is the fundamental vibration period of most buildings in the area (i.e., 2-3 storey buildings). Moreover, 0.3 s is compatible with the results of modelling provided by Gaudiosi et al. (2021), Giallini et al. (2020), Grelle et al. (2020) for the same areas.

The map of Fig. 6 shows an output that is in good agreement with the geophysical data (i.e., $V_{S30}$ in Fig. 7) and geomorphological data (i.e., elevation and slope in Figs. A2 and A3 in Appendix A2 and A3, respectively) and, therefore, highlights local site effects. In fact, referring to Fig. 6, it can be noted that the highest $Sa_{0.3}$ values well describe the valleys' trend (i.e., the largest and continuous Tronto River valley) and the two extended areas in the southern part of the map (i.e., near Petrana and Torrita villages), which are characterized by lowest values of $V_{S30}$ (Mori et al., 2020a). Fig. 8 shows the ShakeMap of $Sa_{0.3}$ regarding the Central Italy earthquake occurred on August 24, 2016 for the same area sketched in Fig. 6. As a general issue referring to the ShakeMaps, the higher the distance from the epicentre (the star in Fig. 8) and the lower the predicted $Sa_{0.3}$. Hence, the ShakeMaps does not provide ground motion variability induced by the local site condition (i.e., sub-soil setting and topography). In detail, ShakeMap provides $Sa_{0.3}$ equal to 0.36 g for the entire area of Arquata del Tronto (square A in Fig. 8) and equal to 0.99 and 1.08 g for Amatrice (square B in Fig. 8).

Referring to A and B close-ups of Fig. 6, Fig. 9 shows the mean values of $Sa_{0.3}$ in the left side and the standard deviation, std, values in the right side. It should be noted that the uncertainty is provided by a combination of the input data values. The uncertainty increases referring to input data values for which the ML is not well trained (Figs. 2 and 3 and discussion in § 2). For instance, std values around 0.3-0.4 are in the areas of inhabited villages, characterised by input data values widely represented in the training dataset, while values in the range 0.6-0.8 are observed in correspondence with the combination of high slope values and high $V_{S30}$ values, which are underrepresented in the training dataset.

In addition to the maps, Fig. 10 shows the profiles (2 at Amatrice and 1 at Arquata del Tronto) of Sa at 0.3 s and the comparison with the values of the same shaking parameter, calculated with different methodological approaches: ground motion prediction with ML approach (this study), 2D numerical simulations (modified after Gaudiosi et al., 2021; Giallini et al., 2020; Grelle et al., 2020), and ShakeMap (2021). All the models are defined for the geometric mean (geoH) of the horizontal components. As ShakeMaps are released for the maximum of the horizonal components, the ShakeMap values are converted to geoH according to the empirical relation proposed by Beyer and Bommer (2006). The three profiles were chosen because they represent three very different geological and geomorphological structures: narrow valley (section AA' in Fig. 10, Arquata del Tronto), plateau

of soft ground (section BB' in Fig. 10, Amatrice), morphology of a mountain peak with coverage of soft ground (section CC'
in Fig. 10, close to Amatrice). As a matter of fact, the adopted ML approach reproduces the so-called valley effect, as in the
case of Arquata del Tronto shallow valley (see the trend for $200 \leq x \leq 400$ m in AA'), the combined lithostratigraphic and
topographic effects, as in the case of Amatrice village (see the trend for $200 \leq x \leq 500$ m in BB'), and the topographic
amplification, as in the case of the AMT accelerometric station (see the trend for $100 \leq x \leq 200$ m in CC'). It should be noted
that the trend of the values of our study reproduces that of the numerical simulations, also getting closer to the recorded values
at Osservatorio Sismico delle Strutture (OSS, a network of buildings and bridges monitored *in continuum* by the Italian Civil
Protection Department) site and AMT station (Luzi et al., 2019; stars in BB' and CC'). Moreover, the profiles provided by the
ML approach are much more articulated and complex than the constant value (horizontal dashed line) of the ShakeMap, which
obviously fails to grasp the local site effects at this scale. The difference between different methodology and the recorded
values were quantified according to the following Equation and are provided in Table 5.

$$\varepsilon_{Sa} = \frac{Sa_{0.3 \text{ estimated}} - Sa_{0.3 \text{ recorded}}}{Sa_{0.3 \text{ recorded}}} \cdot 100 \qquad (2)$$


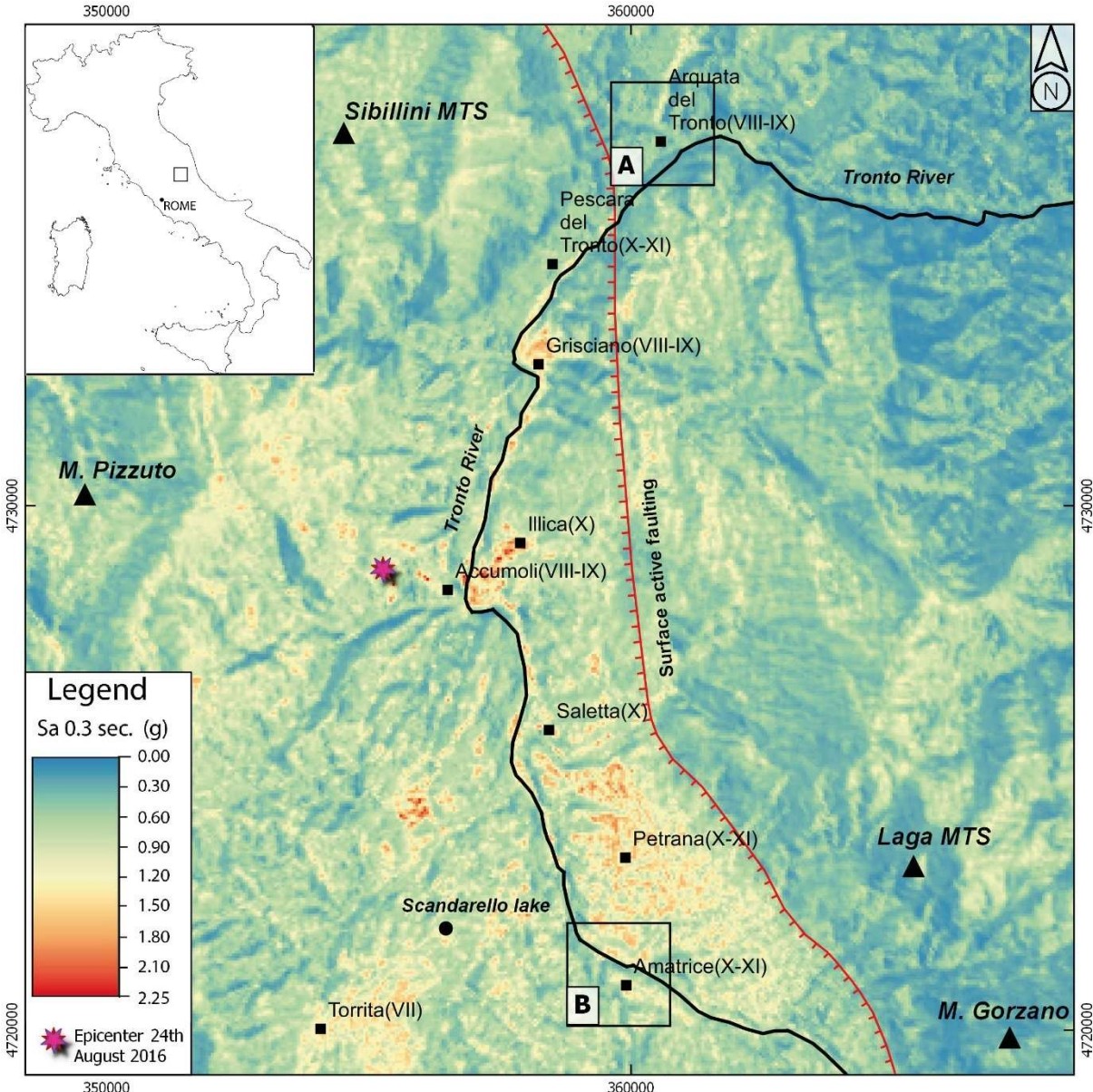

Figure 6. Ground motion prediction map of Sa$_{0.3}$ (resolution 50 m x 50 m) regarding the Central Italy earthquake occurred on August 24, 2016. I_MCS values retrieved by Galli et al. (2017) are reported next to the name of the villages. A and B squares are referred to the close ups at Arquata del Tronto and Amatrice, respectively. The surface active faulting, sketched in the figure, has been slightly modified after Galli et al. (2017).


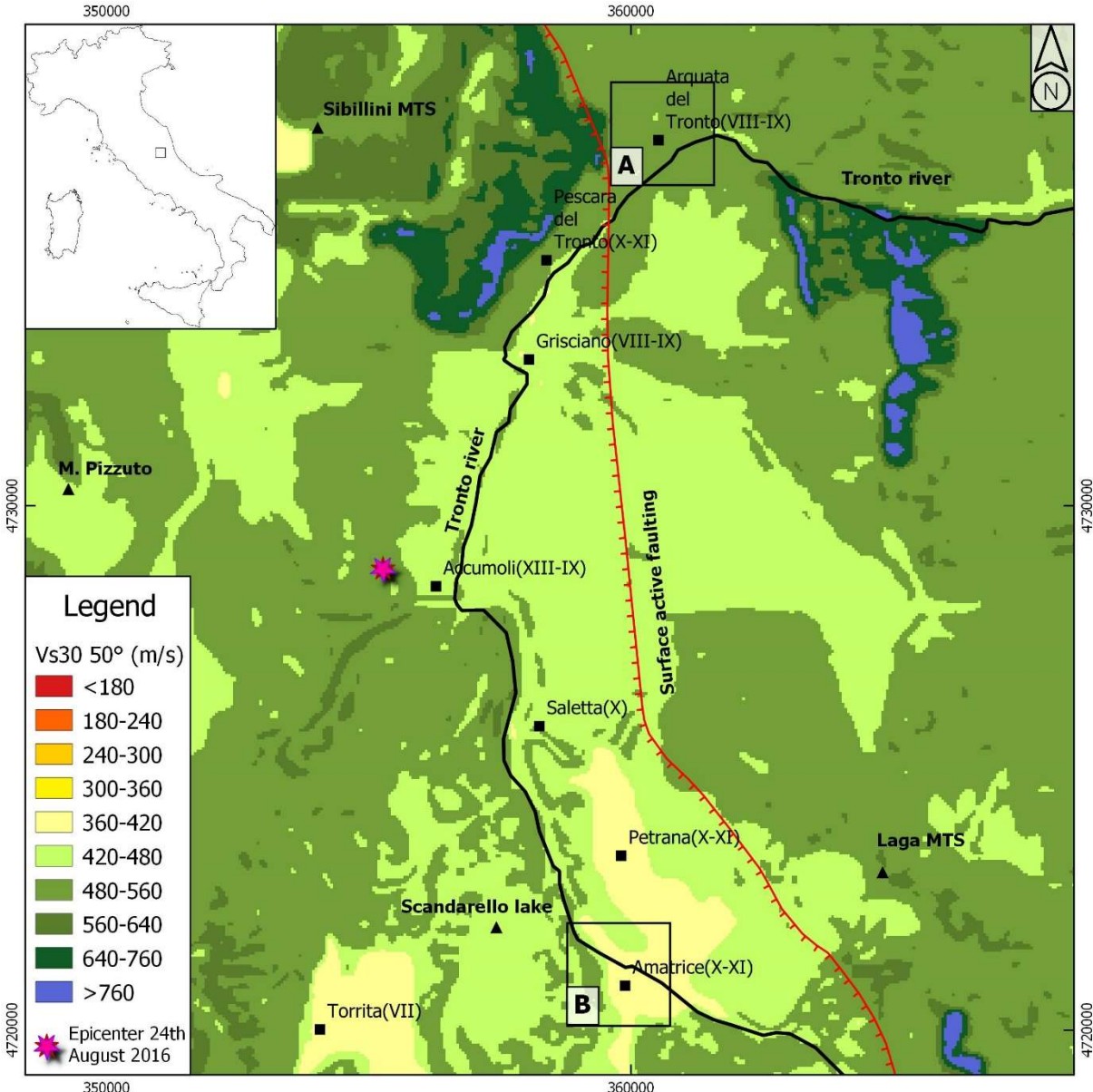

Figure 7. V$_{S30}$ maps for the area of interest shown in Fig. 6. It can be noted that two extended areas in the southern part of the map (i.e., near Petrana and Torrita villages) are characterized by lowest values of V$_{S30}$ inducing the highest Sa$_{0.3}$ values (i.e., valley effect) as shown in Fig. 6.

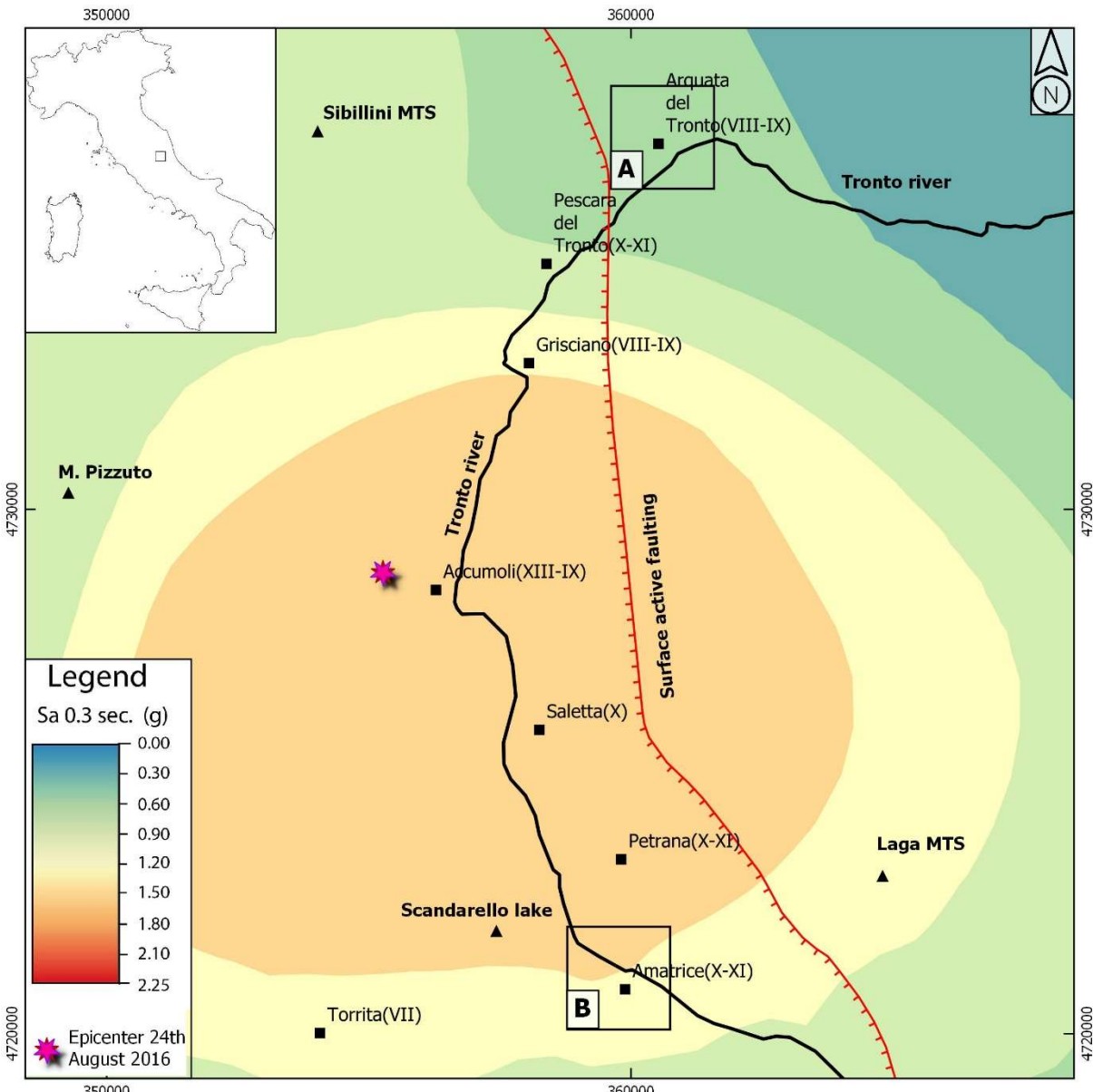


**Figure 8. ShakeMap (slightly modified from ShakeMap, 2021) of Sa$_{0.3}$ regarding the Central Italy earthquake occurred on August**
**24, 2016. A and B squares are referred to the close-ups at Arquata del Tronto and Amatrice, respectively. From the centre of the**
**figure to the border, the homogenous coloured areas correspond to 1.20-1.50 g, 0.90-1.20 g, 0.60-0.90 g, 0.30-0.60 g, and 0.01-0.30 g**
**intervals. It is evident that the map does not capture the variability at short distances**

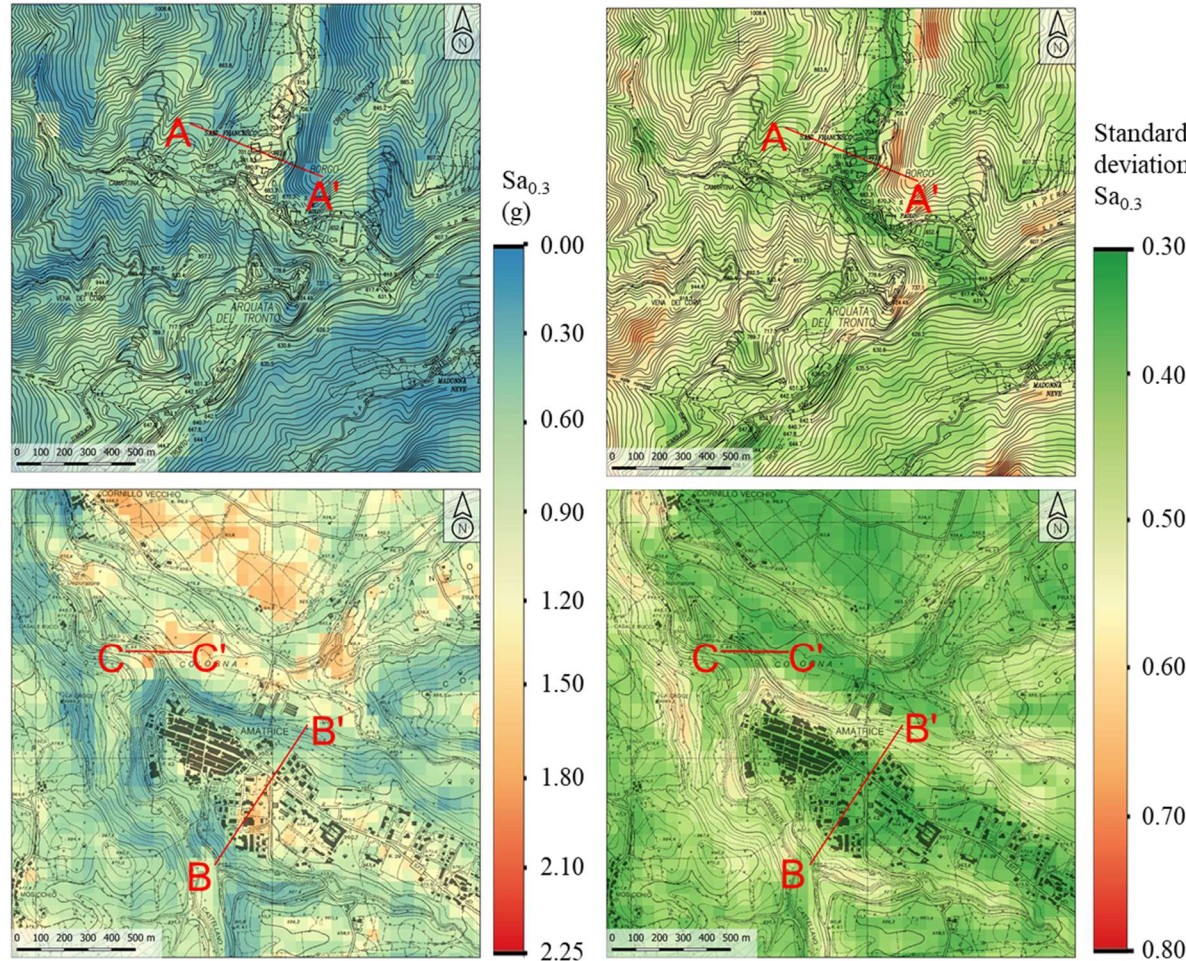


**Figure 9. Ground motion prediction maps (Central Italy earthquake occurred on August 24, 2016) regarding the Arquata del**
**Tronto (top) and Amatrice (bottom) in terms of Sa$_{0.3}$ mean value (left) and standard deviation (right) (resolution 50 m x 50 m). The**
**base topographic layer was retrieved from Regione Marche (2021) and Regione Lazio (2021) for Arquata del Tronto and Amatrice**
**The uncertainty estimation is available here: https://it.mathworks.com/help/stats/gaussian-process-regression-models.html.**


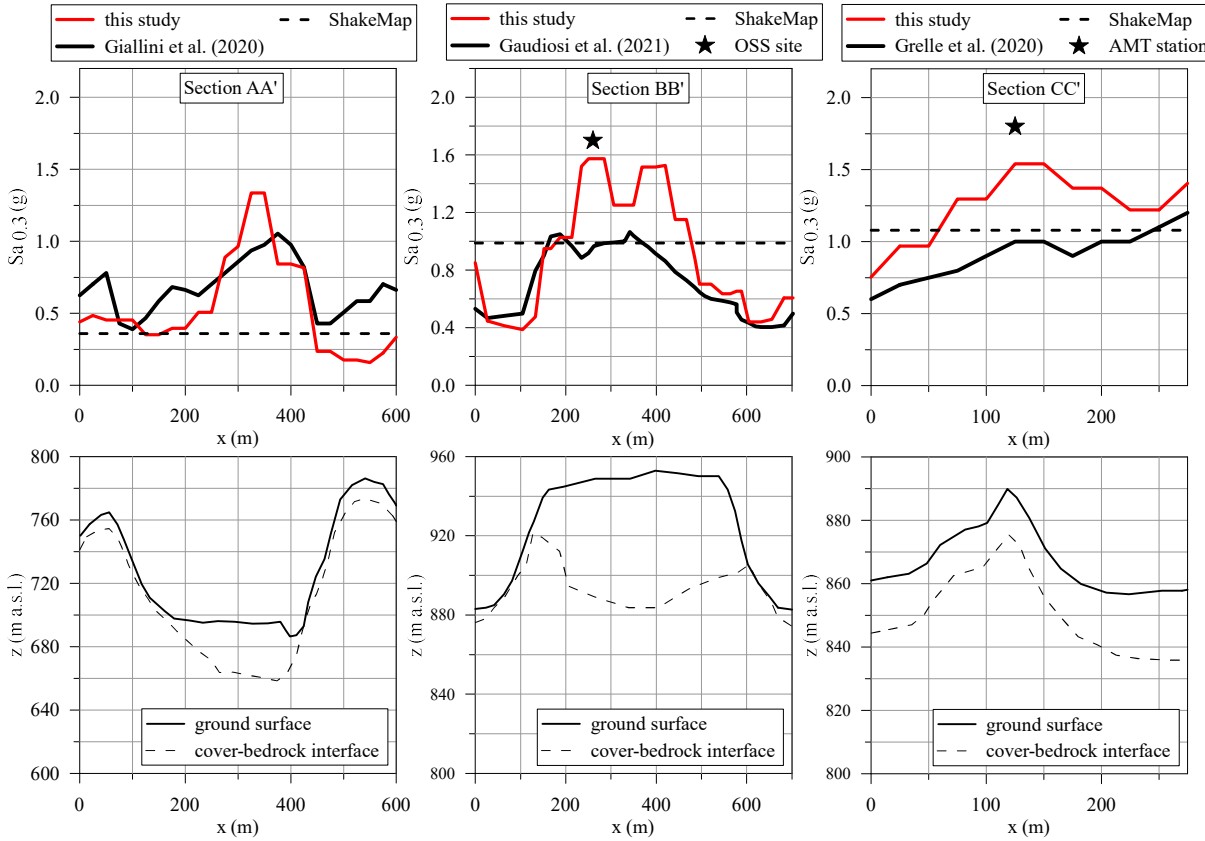

**Figure 10. Profiles of Sa$_{0.3}$ (top) for Central Italy earthquake occurred on August 24, 2016 and simplified sub-soil sections (bottom) of Arquata del Tronto (Section AA') and Amatrice (Sections BB' and CC'). Cross sections' locations are in Fig. 9. Sa$_{0.3}$ profiles and geological information retrieved and modified after Gaudiosi et al. (2021), Giallini et al. (2020), Grelle et al. (2020); ShakeMap (2021). The black stars indicate values recorded at the OSS site and AMT station (for details see the text).**


**Table 5. Difference ε in percentage between Sa$_{0.3}$ determined by means of different methodologies and recorded values for Central Italy earthquake occurred on August 24, 2016.**

|  | Section BB' | Section CC' |
|---|---|---|
|  | $\varepsilon_{Sa}$ (%) | $\varepsilon_{Sa}$ (%) |
| this study | -7 | -14 |
| Numerical | -43 | -44 |
| ShakeMap | -42 | -40 |


**Discussion and conclusions**


Intensity and frequency contents of ground motions can be altered by many factors. Up until now, numerous empirical models
of ground motion amplification have been developed based on conventional regression analyses, considering few key factors
such as intensity measures of rock motions, shear wave velocities of soils, and territory morphology. Since Machine Learning
techniques have been applied to many fields, this work investigated on efficacy of using such techniques for developing models
to predict ground motion over large areas with a 50 m resolution raster.
A set of about 16,000 ground motion data from Italian and European networks were adopted to train a Gaussian Process
Regression model, while recordings by 241 stations of the seismic events occurred in Italy on October 30, 2016 were used to
test the same model. Peak ground acceleration and velocity, and spectral acceleration at 3 periods (i.e., 0.3, 1, and 3 s) were
compared to the recorded data allowing to obtain residuals. With reference to the training dataset, mean value and standard
deviation of the residuals' distribution were found equal to about 0 and to about 0.1, respectively. With reference to the test
dataset characterised by magnitude equal to 6.5, mean value and standard deviation of the residuals' distribution were found
equal to 0.01 and 0.3, respectively. Hence, the performance of the adopted Machine Learning technique was confirmed
satisfactory also for magnitude higher than 6.
In addition, maps of ground motion in terms of peak ground acceleration, peak ground velocity, and of spectral acceleration at
the selected three periods were produced for the Central Italy seismic event occurred on August 24, 2016. Profiles of intensity
measures were in satisfactory agreement with those obtained by means of advanced numerical simulations of seismic site
response referring to the same seismic event. Moreover, the adopted Machine Learning approach greatly improves the
performance of existing methods for the analysed case studies.
Three main novelties of the work are synthesized in the following:
1) forecast of ground motion with high resolution (i.e., a 50 m x 50 m raster), in agreement with results of local scale numerical
modelling. This outcome is achieved by means of Machine Learning techniques and large datasets including
morphological, geological, geophysical, and geotechnical features (mainly the seismic microzonation dataset; DPC, 2021).
Moreover, about 1,000 seismic events recorded by 1,435 accelerometric stations (ESM, 2021; ITACA, 2021) were
analysed. The Machine Learning approach combines morphological and subsurface proxies: elevation, first and second
order topographic gradient (define the morphological characteristics of the territory), mean shear wave velocity in the upper
30 m (defines the dynamic response of a site as induced by the subsoil condition). Magnitude, epicentral, and hypocentral
distances provide the source conditions;
2) use of robust statistical techniques such as Gaussian Process Regression. Among the machine learning based models, the
model developed by the regression and Gaussian approach provides the best performance in terms of both precision and
accuracy, that are standard deviation and mean value of the residuals' distribution, respectively.
In a nutshell, the novelty of this work is the use of the Machine Learning approach based on the analysis of a huge database of
geological, geophysical, and geotechnical data, built with Seismic Microzonation studies for the entire Italian territory. The
quality and quantity of this database allow a robust application of Machine Learning including the prediction of local site
effects (i.e., lithostratigraphic and morphological) on the seismic ground motion.
In terms of applications, the ground motion maps generated by means of the proposed Machine Learning approach are useful
both for urban planning (aimed at reducing seismic risk) and for emergency management (aimed at a *near real time* estimation
of shaking scenarios). With reference to the emergency phase, by knowing the position and depth of the hypocentre and the
magnitude of the event (in Italy these data are available a few minutes after the event), it is possible producing ground motion
maps in near real time. Overall, considering that the paradigm should be shifted from managing disasters to managing risk, the
proposed methodology could represent a key-tool in seismic risk mitigation strategies deployed both pre and post seismic
event.
Evaluation of the spatial correlation structure was studied to provide the relation between local site effects and spatial resolution
of ground motion maps; results of such analysis were not reported in the main text since it is out of the scope while preliminary
results in terms of sill and range are reported in the Appendix A4 referring to the seismic event occurred on October 30, 2016
(i.e., the strongest of the Central Italy seismic sequence).
In conclusion, the research on this topic will continue and focus on specific goals, which are listed in the following:
- improve the method with more input proxies, made available after the seismic microzonation project for the whole national
territory. In detail, maps of the depth to the engineering bedrock and of the fundamental frequency of the deposit will be soon
available and allow to use such parameters as input data for the Machine Learning approach;
- improve the method with worldwide seismological dataset;
- improve the spatial resolution of existing input proxies integrating remote sensing data;
- improve the spatial correlation analysis.
**Author contributions**
Conceptualization: FM, GA. Data curation: FM, AM, RS. Formal analysis: FM, RS. Funding acquisition: MM. Methodology:
FM, AM, GF, GA, RS, MM, GN. Project administration: MM. Supervision: FM, MM, GN. Validation: FM, AM, GF, GA.
Visualization: AM, GF. Writing – original draft preparation: FM, GF, GN. Writing – review & editing: FM, AM, GF, GA,
RS, MM, GN.
**Competing interests**
The authors declare that they have no conflict of interest.

## Acknowledgements

Authors would like to thank F. Bramerini, S. Castenetto, A. Gorini and D. Spina, (Italian Department for Civil Protection), for the useful discussions. We also thank S. Giallini and I. Gaudiosi (both from CNR IGAG, Italy) for providing the ground motion data obtained by means of numerical simulation for Amatrice and Arquata del Tronto areas (Italy).

## Financial support

This research was supported by the Presidency of the Council of Ministers, Italian Department for Civil Protection, in the framework of the project "Contratto concernente l'affidamento di servizi per il programma per il supporto al rafforzamento della Governance in materia di riduzione del rischio sismico e vulcanico ai fini di protezione civile nell'ambito del PON Governance e Capacità Istituzionale 2014–2020 - CIG6980737E65" (M. Moscatelli scientific coordinator for CNR).

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

**Appendix A1. Fig. A1 - The contribution of each proxy to the total reduction in the standard deviation of the**
**residuals for the PGA**.

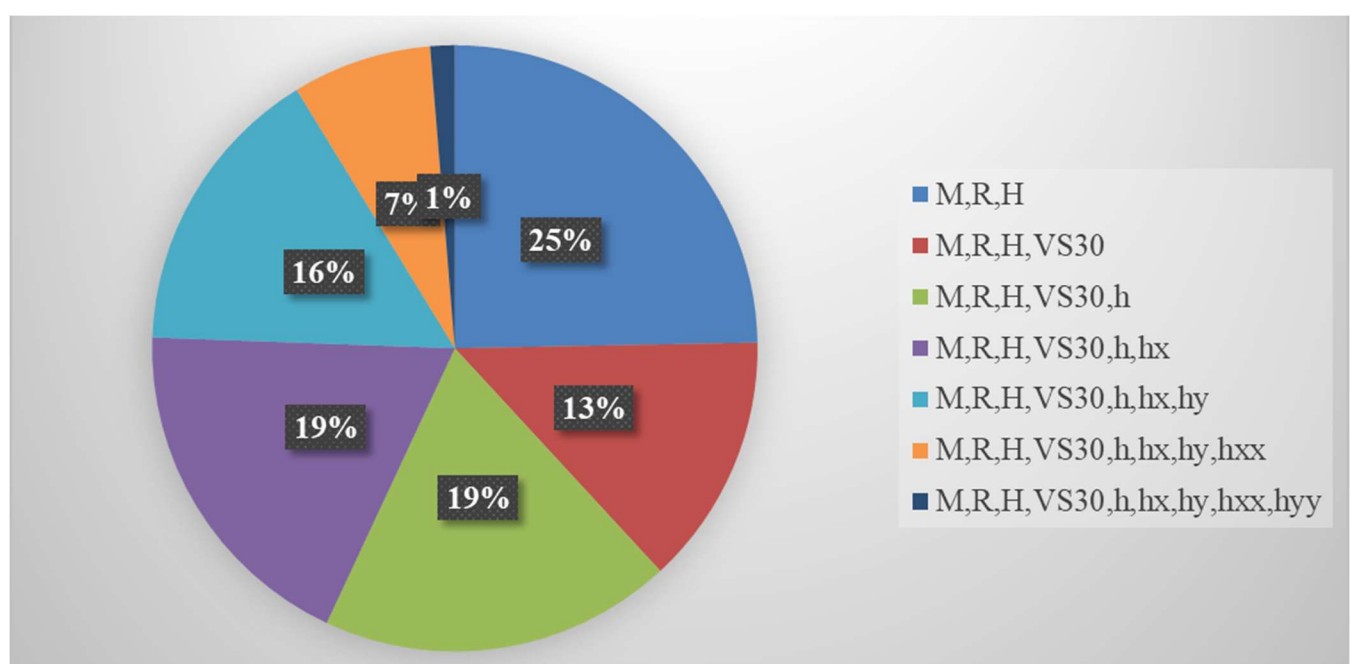






**Appendix A2. Fig. A2 - Elevation map for the area of Figs. 6-8 in the manuscript**

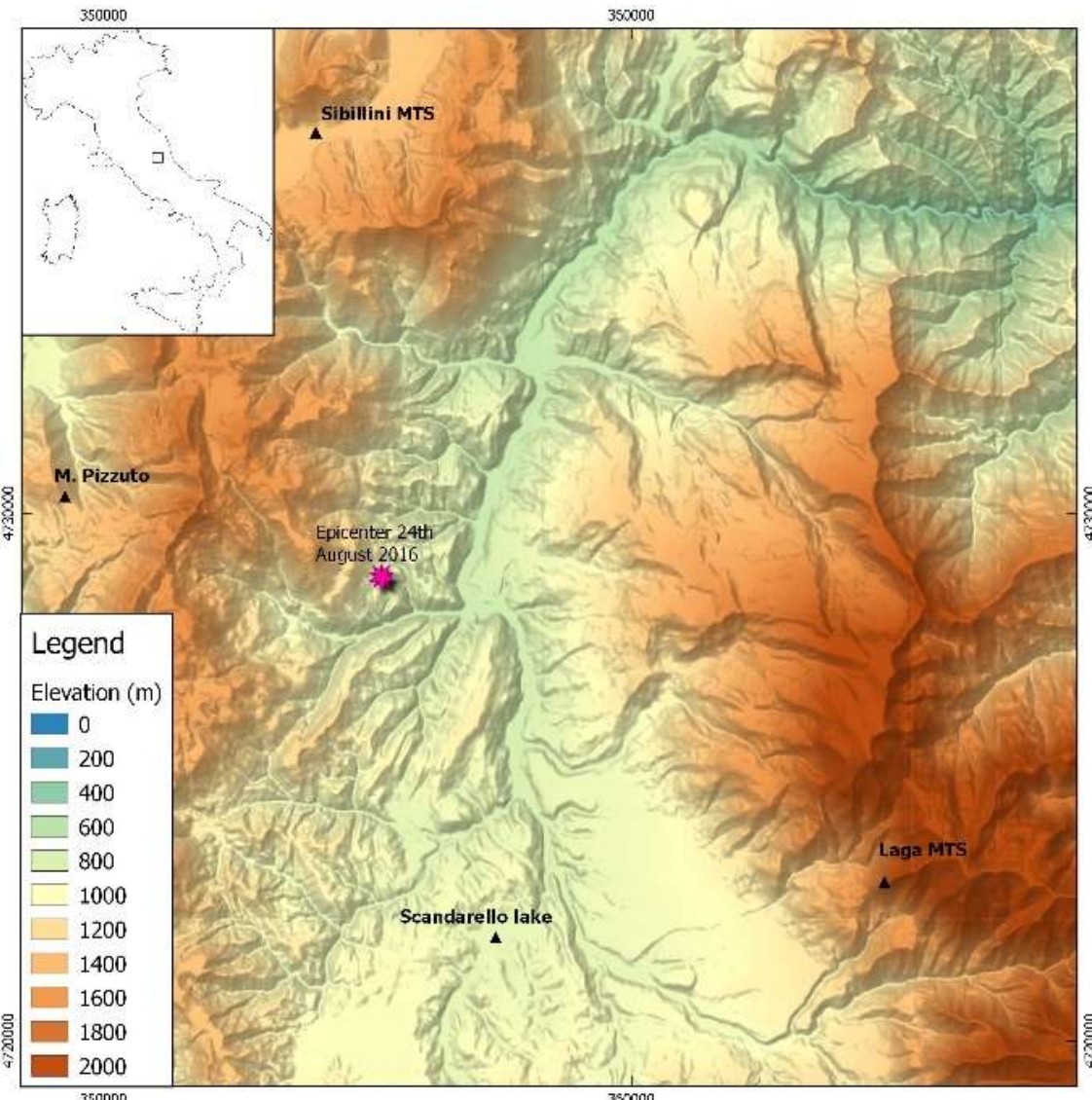






**Appendix A3. Fig. A3 - Slope map for the area of Figs. 6-8 in the manuscript**

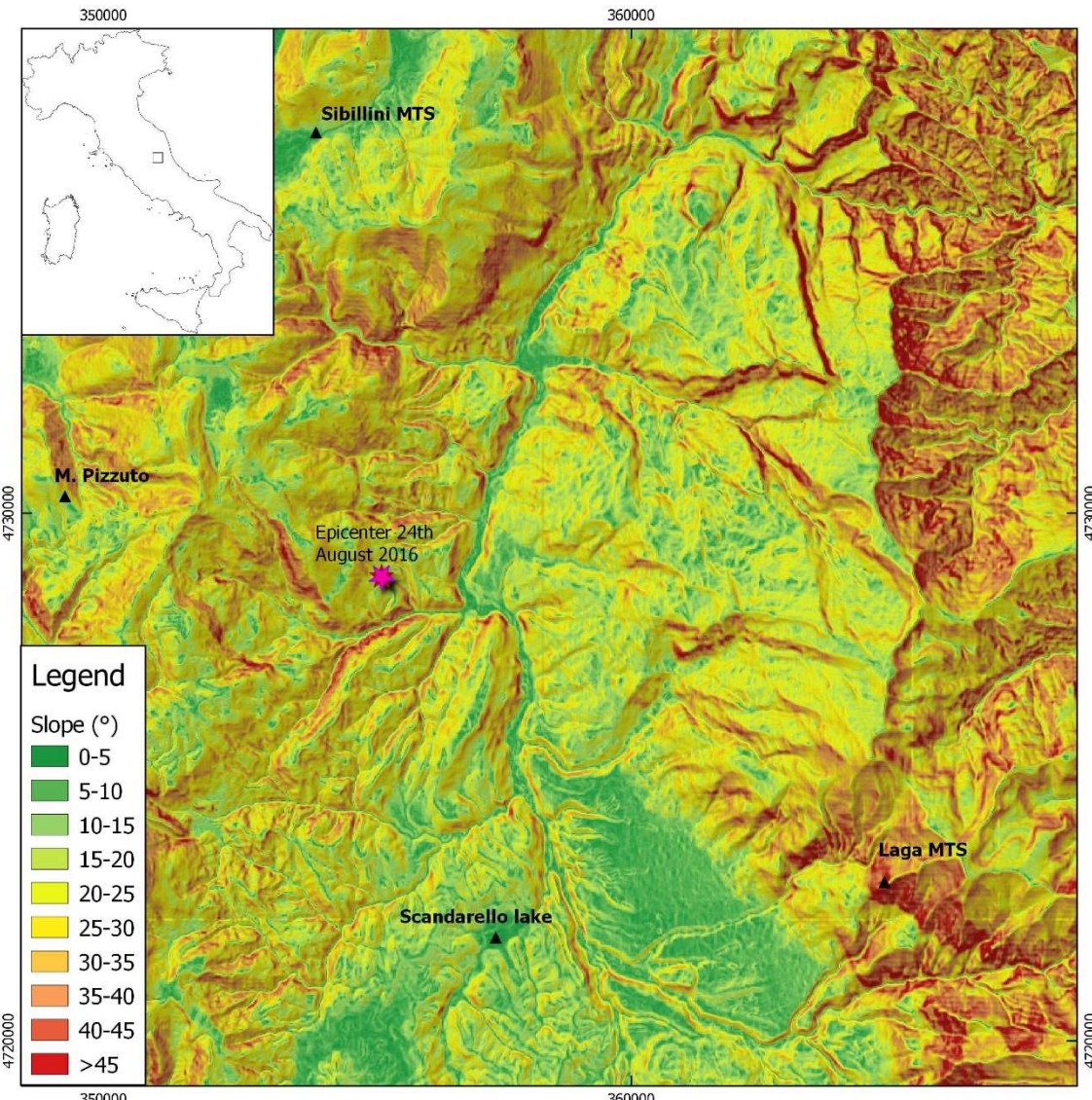





## Appendix A4. Spatial correlation structure of the predicted maps

In this appendix we want to preliminarily deal with the spatial correlation of the IM parameters. In fact, the spatial correlation of ground-motion IMs represents a key issue in the seismic risk assessment, particularly in loss analysis (Infantino et al., 2021; Schiappapietra et al., 2020, 2021). The geostatistical tool widely adopted to analyse the spatial correlation of geological and geotechnical data (Paolella et al., 2021, Raspa et al., 2008, Salvatore et al., 2019, Spacagna et al., 2018) is the semi-variogram (Chilès and Delfiner, 2012). The spatial structure is evaluated by assessing the dissimilarity of the variables measured at different locations. First, referring to the variable of interest (in this case, one of the selected IMs), the experimental semi-variogram $\hat{\gamma}(\boldsymbol{h})$ is calculated from data using the method of moments (Chilès and Delfiner, 2012):

$$\hat{\gamma}(\boldsymbol{h}) = \frac{1}{2n(\boldsymbol{h})} \sum_{i=1}^{n(\boldsymbol{h})} \left\{ z(\boldsymbol{x}_i) - z(\boldsymbol{x}_i + \boldsymbol{h}) \right\}^2 \tag{A1}$$

where $z(\boldsymbol{x}_i)$ and $z(\boldsymbol{x}_i + \boldsymbol{h})$ are the observed values of the variable $z$ (i.e., one of the selected IMs) at the location $\boldsymbol{x}_i$ and $\boldsymbol{x}_i + \boldsymbol{h}$ separated by $\boldsymbol{h}$, and n($\boldsymbol{h}$) is the number of pairs at lag $\boldsymbol{h}$. Under the assumption of second-order stationary, the semi-variogram increases with $\boldsymbol{h}$ up to a constant value of $\hat{\gamma}(\boldsymbol{h})$. In this study, to assess the spatial structure of the variables (predicted IMs), the experimental variogram estimated from the predicted maps is fitted with the best fit model (i.e., the exponential model):

$$\gamma(h) = C \left[ 1 - \exp\left( \frac{-3h}{a} \right) \right] \tag{A2}$$

where the parameters $a$ and $C$ are called respectively range and sill. The range defines the correlation distance, namely, the separation distance at which the data are spatially independent, and the sill represents the variance of the random process, limit value of $\gamma(\boldsymbol{h})$.

For the Central Italy event occurred on October 30, 2016 and for all the predicted IMs maps (i.e., PGA, $Sa_{0.3}$, $Sa_1$; see Fig. 6 and supplementary materials), the spatial structure was performed with the GSTAT package (Pebesma, 2004) of the R software (R Core Team, 2021). The IMs values were extracted from the predicted maps with a regular punctual grid of 50 m x 50 m. The isotropic experimental semi-variograms were computed and fitted with the above-mentioned exponential model. As an example, Fig. A4 shows the semi-variogram of the predicted $Sa_{0.3}$ map. The spatial structure of all predicted IMs maps was characterized by the nested exponential model. The nested variograms highlight the presence of a double structure at different scales, i.e., a short-scale and a long-scale variability.

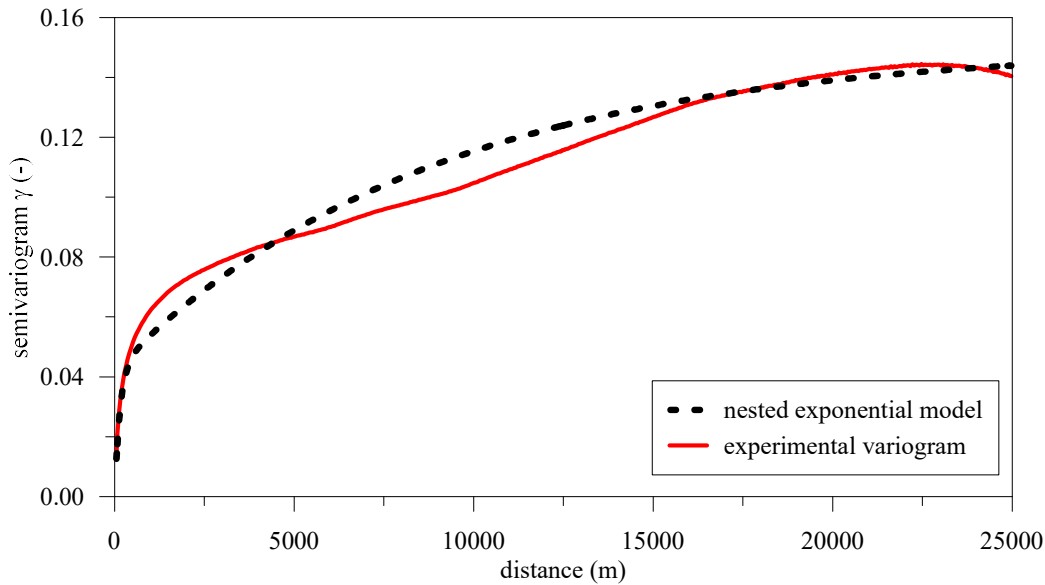

**Figure A4. Semi-variogram of the predicted Sa$_{0.3}$ map (Central Italy event occurred on October 30, 2016): experimental variogram based on the adopted ML approach and best-fitting model (nested exponential).**

In this case, two ranges and two sills are obtained for two levels of variability. Table A1 shows the sill and range values for the nested exponential models of all predicted IM maps. The first range, or short-scale structure, captures the first source of variability (first sill) over hundreds of meters induced by lithostratigraphic site conditions and morphological variability. The long-scale structure captures the variability over thousands of meters and could be referred to regional geological units and large-scale morphological features. Furthermore, a significant part of the variance, around 30-40% of the total, are captured at short-scale.

An exhaustive treatment of this topic is beyond the scope of this work. We are now studying the spatial variability of input parameters that contribute to generate the target IM maps, and this will be the subject of a future paper. By the way, the preliminary results enlighten the importance to generate ground motion prediction maps with a spatial resolution in the order of hundreds of meters, to improve their quality in terms of predictivity. Seismic hazard maps should also include these specifications to consider the short-scale effects, even if starting from basic hazard maps with a resolution in the order of 2-5 km.

**Table A1. Sill and range values of the nested exponential model for all the predicted IM maps.**

| IM | Short-scale structure | | Large-scale structure | |
|---|---|---|---|---|
| | sill | range [m] | sill | range [m] |
| PGA | 0.01080 | 600 | 0.022550 | 28500 |
| Sa$_{03}$ | 0.04250 | 450 | 0.108000 | 26700 |
| Sa$_1$ | 0.00530 | 450 | 0.010500 | 21600 |
| Sa$_3$ | 0.00022 | 750 | 0.000265 | 20400 |