# Peer review of "Ground motion prediction maps using seismic microzonation data and machine learning"

_Natural Hazards and Earth System Sciences, 2021_

## Referee Comment (RC2)

This document presents a review of the manuscript titled as "Ground motion prediction maps using seismic microzonation data and machine learning" by Mori et al. The main objective of the work is to apply the machine learning approach to provide ground motion maps predicted based on stratigraphic and morphological parameters. This is a somewhat novel idea when the validity of the outcome could be well-established.

The use of English language is generally fine. However, there are repetitive misuses of some particular words/phrases that make the reading tedious and confusing. There are also some misuses of punctuations (e.g. comma). Some examples have been given below. This reviewer strongly recommends a proofread of the article by a better English user.

The organization of the article is generally fine. However, the abstract does not seem to be a good summary of the work. It gives us the impression that the current work deals with near-real-time prediction of damage, which is not the case. The last sentence refers to a conclusion regarding the effects of short-distance variability, which has not been extensively addressed or proven by this article. The introduction segment could be improved. The texts between Line no. (L) 74 and 95 seem irrelevant. They should be presented in a different segment under a different title. The major weakness of this article is that the method section remains completely underdeveloped. As machine learning is at the core of this work, the method of evaluating the training dataset and validating the outputs should be clearly presented. At the current state, reproducibility of the results of this work remains under doubt. In the following sections, the interpretation of results could also be improved. The last segment related to the spatial correlation of the ground motion predictions seems completely underdeveloped. If the authors intend to present it in a future article (as mentioned in L 348-349), then there is no point addressing this issue and drawing some concrete conclusions without any explanation within this article.

Therefore, this reviewer recommends a major improvement of the article before being considered for the publication. The main comments are as below:

**Scientific Comments:**

- The presentation of the datasets as maps would be helpful. Median and percentiles could be indicated on Figure 1 and 2 for a better comprehension.
- The method section must be improved. How are the outputs estimated from the input data. Which parameters have been used and how? Which proxy performed the best? What are the differences among different ML prediction models (Table 2) and why have they been adopted? How come the SD values are reduced by 45-60% (L217)? It's not clear how Openquake was used to determine the IMs – which site and fault parameters have been used for the calculation?
- The reviewer disagrees with the statement in L 264-365. It's very difficult to visually determine if the maps in Figure 4 and 5 are coherent with geological and geomorphological

characteristics. No such maps of the area have been presented. For comparison, the existing ground motion estimation for this area (e.g. from the shake map) and/or microzonation map could be presented on a same color scale.

- The estimation of uncertainty presented in Figure 5 should be elaborated. What kinds of uncertainties have been considered and how have they been estimated?
- The validation of the results seems insufficient. It uses the results from only one earthquake, for one spectral period and only 3 cross-sections. How are the Sa profiles (in Figure 6) obtained and modified from Giallini et al.? What about the ground motions at other spectral periods? Do they compare well? Other than these 3 cross-sections from two sites, how many more sites have been used to validate the results? How long are the cross-sections?
- Segment 4.2 should be either eliminated or better explained. Results shown in Figure 7 and Table 5 should be interpreted in plain terms. The fact that the estimated short-scale variability is related to site conditions should be validated in terms of physical parameters. Which input parameters are causing these variations and to what extent? How close it is to the reality (example from a site)?

**Editorial Comments:**

- The phrases 'referred to' or 'with reference to' have been heavily repeated and misused. All of them must be replaced by appropriate vocabulary or the sentences should be rewritten. The phrase 'bearing in mind' has been tediously repeated.
- The resolution of the map mentioned as '50 X 50 m' should be corrected as '50 m X 50 m' for the entire article.
- For all the large numbers used in the article, the thousand separator should be corrected fro 'XXX'000' to 'XXX,000' In English, comma separates the thousand.
- Figure 2 has not been mentioned or explained anywhere. The cross-section labels on Figure 5 are not readable. The long web-address in the title could be replaced by a reference. The Figure 6 title should mention that the cross-sections are presented from Figure 5.
- L 108-112 should be rewritten. The current form is grammatically incoherent.
- In L 150 the phrase 'here adopted' should be corrected as 'adopted here'.
- In L 232 'gauge the IMs' sounds unnatural. It could be replaced by 'estimate the IMs'.
- The webpage links should be referred to as a reference with last access date. It is sufficient to cite them as references in the text rather than citing the link everywhere.
- L 376 and 385 are not expressed as full sentences. Either they should be turned into paragraph subtitles by using a colon (:) after 'following' in L375 or they should be re-written as full sentences. Similar remark for L 397-403.

---

## Author Comment (AC1)

**Title: "Ground motion prediction maps using seismic microzonation data and machine learning" by Federico Mori et al., Nat. Hazards Earth Syst. Sci. Discuss.,**

Dear Referee,

we thank you for your thorough assessment of our paper. We have carefully addressed the comments and made corresponding changes to the manuscript. We have modified and added some figures. We have carefully revised the ambiguous statement in the article.

**Referee comment RC1 https://doi.org/10.5194/nhess-2021-282-RC1**

The manuscript utilizes machine learning in ground motion prediction and compares ML based techniques with GMM and ShakeMap. The former has been shown to have better performance than the latter two. I am not surprised by the results, as has been demonstrated by many that ML techniques are advantages over parametric GMMs. ShakeMap is basically also based on GMMs which reply on an input grid-based  $V_{S30}$  map.

General comments:

• What is the logic behind the selection of site proxies? For instance, why do you utilize elevation? Is there any physical reasoning? Is it necessary to use elevation, slope (hx and hy) and curvature (hxx and hyy) simultaneously? Could you provide a plot or some discussions on the performance of each site proxy in the best performing model (GPR)? I recommend the authors expend a bit on the performance of these site proxies?

**Reply:**

Zhou et al. (2020) suggests that the parameters of the topographic elevation, the first gradient of the elevation and the second-order gradient in two orthogonal directions are enough to provide the acceptable topographic effect model.

So, the philosophy was to add these five proxies (topographic effect proxies) respect to the four standard proxies (magnitude, epicentral distance, hypocentral depth,  $V_{s30}$ ) already used in the Ground Motion Models GMMs.

In the case of PGA, for example, these are the differences in terms of performance (14% reduction in terms or Root mean squared error, RMSE, and 17% reduction in terms of Mean Absolute Error, MAE). In terms of residuals, there is a 46% reduction of standard deviation.

| Selected proxies | Root Mean Squared
Error (RMSE) | Mean           |           | mean of   | standard     |
|------------------|-----------------------------------|----------------|-----------|-----------|--------------|
|                  |                                   | Absolute Error | R-squared | residuals | deviation of |
|                  |                                   | (MAE)          |           |           | residuals    |
| GMMs standard 4  | 0.37                              | 0.29           | 0.85      | -0.001    | 0.3          |
| proxies          |                                   |                |           |           |              |
| This study 9     | 0.31                              | 0.24           | 0.89      | -0.000033 | 0.161        |
| proxies          |                                   |                |           |           |              |

Specific comments:

• Line 74-95: these paragraphs have many details. I suggest they better be moved to other sections, rather than in the introduction. The introduction shall serve to intrigue the readers to read the paper, but these paragraphs are too detailed and may be counterproductive.

**Reply:**

The lines 74-95 were rephrased and moved at the beginning of the section 3.2 "Testing phase".

---

## Author Response (AR1)

**Consiglio Nazionale delle Ricerche (CNR)**
**Istituto di Geologia Ambientale e Geoingegneria (IGAG)**

**Title: "Ground motion prediction maps using seismic microzonation data and machine learning" by Federico Mori et al., Nat. Hazards Earth Syst. Sci. Discuss.,**

Dear Editor,

we thank you and reviewers for their thorough assessment of our paper. We have carefully addressed the reviewers' comments and made corresponding changes to the manuscript. We have modified and added some figures. We have carefully revised the ambiguous statement in the article. The response to reviewers' comments has been reported below.

**Referee comment RC1** https://doi.org/10.5194/nhess-2021-282-RC1

The authors would like to thank the reviewer for his comments.

The manuscript utilizes machine learning in ground motion prediction and compares ML based techniques with GMM and ShakeMap. The former has been shown to have better performance than the latter two. I am not surprised by the results, as has been demonstrated by many that ML techniques are advantages over parametric GMMs. ShakeMap is basically also based on GMMs which reply on an input grid-based $V_{S30}$ map.

General comments:

- What is the logic behind the selection of site proxies? For instance, why do you utilize elevation? Is there any physical reasoning? Is it necessary to use elevation, slope (hx and hy) and curvature (hxx and hyy) simultaneously? Could you provide a plot or some discussions on the performance of each site proxy in the best performing model (GPR)? I recommend the authors expend a bit on the performance of these site proxies?

Reply:

Zhou et al. (2020) suggests that the parameters of the topographic elevation, the first gradient of the elevation and the second-order gradient in two orthogonal directions are enough to provide the acceptable topographic effect model.

So, the philosophy was to add these five proxies (topographic effect proxies) respect to the four standard proxies (magnitude, epicentral distance, hypocentral depth, $V_{s30}$) already used in the Ground Motion Models GMMs.

In the case of PGA, for example, adding these five proxies to the four standard proxies, performance improves (14% reduction in terms or Root mean squared error, RMSE, and 17% reduction in terms of Mean Absolute Error, MAE). In terms of residuals, there is a 46% reduction of standard deviation.

This table shows the details of the performance improvement.

| Selected proxies | Root Mean Squared Error (RMSE) | Mean Absolute Error (MAE) | R-squared | mean of residuals | standard deviation of residuals |
|---|---|---|---|---|---|
| GMMs standard 4 proxies | 0.37 | 0.29 | 0.85 | -0.001 | 0.3 |
| This study 9 proxies | 0.31 | 0.24 | 0.89 | -0.000033 | 0.161 |

Specific comments:

- Line 74-95: these paragraphs have many details. I suggest they better be moved to other sections, rather than in the introduction. The introduction shall serve to intrigue the readers to read the paper, but these paragraphs are too detailed and may be counterproductive.

Reply:

The lines 74-95 were rephrased and moved at the beginning of the section 3.2 "Testing phase".

[Figure]

**Consiglio Nazionale delle Ricerche (CNR)**
**Istituto di Geologia Ambientale e Geoingegneria (IGAG)**

**Referee comment RC2** https://doi.org/10.5194/nhess-2021-282-RC2

The authors would like to thank the reviewer for his comments.

This document presents a review of the manuscript titled as "Ground motion prediction maps using seismic microzonation data and machine learning" by Mori et al. The main objective of the work is to apply the machine learning approach to provide ground motion maps predicted based on stratigraphic and morphological parameters. This is a somewhat novel idea when the validity of the outcome could be well-established.

The use of English language is generally fine. However, there are repetitive misuses of some particular words/phrases that make the reading tedious and confusing. There are also some misuses of punctuations (e.g. comma). Some examples have been given below. This reviewer strongly recommends a proofread of the article by a better English user.

The manuscript was revised accordingly.

The organization of the article is generally fine. However, the abstract does not seem to be a good summary of the work. It gives us the impression that the current work deals with near-realtime prediction of damage, which is not the case. The last sentence refers to a conclusion regarding the effects of short-distance variability, which has not been extensively addressed or proven by this article. The introduction segment could be improved. The texts between Line no. (L) 74 and 95 seem irrelevant. They should be presented in a different segment under a different title. The major weakness of this article is that the method section remains completely underdeveloped. As machine learning is at the core of this work, the method of evaluating the training dataset and validating the outputs should be clearly presented.

Reply to Comment #1.

Abstract and Introduction were modified according to the reviewer suggestion.

"Past seismic events worldwide demonstrated that damage and death toll depend on both the strong ground motion (i.e., source effects) and the local site effects. The variability of earthquake ground motion distribution is caused by local stratigraphic and/or topographic setting and buried morphologies, that can give rise to amplification and resonances with respect to the ground motion expected at the reference site. Therefore, local site conditions can affect an area with damage related to the full collapse or loss in functionality of facilities, roads, pipelines, and other lifelines. To this concern, the *near real time* prediction of ground motion variation over large areas is a crucial issue to support the rescue and operational interventions. A machine learning approach was adopted to produce ground motion prediction maps considering both stratigraphic and morphological conditions. A set of about 16,000 accelometric data and about 46,000 geological and geophysical data were retrieved from Italian and European databases. The intensity measures of interest were estimated based on 9 input proxies. The adopted machine learning regression model (i.e., Gaussian Process Regression) allows to improve both the precision and the accuracy in the estimation of the intensity measures with respect to the available *near real time* predictions methods (i.e., Ground Motion Prediction Equation and shaking maps). In addition, maps with a 50 m x 50 m resolution were

generated providing a ground motion variability in agreement with the results of advanced numerical simulations based on detailed sub-soil models.".

The lines 74-95 were rephrased and moved at the beginning of the section 3.2 "Testing phase".

Section 4.2 "Spatial correlation structure of the predicted maps" was removed from the manuscript and reported as Appendix. The authors would make the readers known about the fact that we are working on the analysis of ground motion spatial correlation and this analysis will be improved in a future work. Hence, the last sentence of the abstract was removed.

The major weakness of this article is that the method section remains completely underdeveloped. As machine learning is at the core of this work, the method of evaluating the training dataset and validating the outputs should be clearly presented.

Figure 4 and the following lines (Line 176 to Line 193) were reported in the revised manuscript (section 3 Method):

"Fig. 4 shows the adopted ML workflow. After having imported and selected the data (input variables and output variables), the training and validation phases begin. In these phases the ML model that will be used is "adapted" or rather the algorithm is adapted to the training dataset. One of the objectives of this phase is the tuning of the model, acting on the hyperparameters (parameters whose value is used to control the learning process) of the algorithm to minimize errors. The k-fold cross-validation technique was used in this work. The models included in "Matlab Regression Learner App" tool have all been tested. The fitting performance (in term of RMSE) on the validation set was considered as an indicator for the generalization ability of models. Among the available models the best fitting performance in terms of RMSE was provided by the GPR model with exponential kernel (Table 2). GPR is a nonparametric, Bayesian approach to regression, which provides uncertainty measurements on the predictions.

The second step is to test the model with the best performance (GPR in this research) adopting a dataset not included in the training and validation phases. The dataset for the 30 October 2016 seismic event was used since the accelerometric data of many accelerometric stations are available. The test is used to evaluate the accuracy of the model in terms of residuals (Eq. 1). In the workflow of Fig. 4 there is also a phase (comparison) that is not part of the standard ML methodology. The comparison with the ground shaking obtained by completely different methodologies was used to further analyse the ML model in terms of ground motion resolution and variability.

Training and cross-validation phases are described in § 3.1. Comparison in terms of residuals with the performance of the existing methods (i.e., an external test) is presented in § 3.2. The comparison with the ground shaking obtained by completely different methodologies is presented in § 4.

Moreover, a detailed description of GPR method is outside the scope of this work. Suggested references for comprehensive descriptions of the GPR method are Rasmussen and Williams (2006) and chapter 6 of MathWorks (2019). The above-mentioned k-fold cross-validation (k=5) method is described in chapter 24 of Mathworks (2019).".

[Figure]

Figure 4. ML workflow adopted in this study. In figure, ESM (ITACA), ALOS, and SM are the open-source datasets from which training and test data for ML models were retrieved.
*The selected test set was the input and output data for the October 30, 2016 seismic event.
** This element of the workflow is not part of the standard ML methodology. This element was introduced to enlighten the capability of the adopted ML procedure in estimating local scale ground motion variability. Comparison against predictions from ShakeMap and 2D numerical simulations was based on August 24, 2016 seismic event input and output data.

At the current state, reproducibility of the results of this work remains under doubt. In the following sections, the interpretation of results could also be improved.

Reply to comment #2.

An Excel file dataset.xls will be attached: input variables are in grey and output variables in yellow.

With this Excel file and the explanation inserted in Line 176 to Line 193 and relative Figure 4, it is possible to reproduce what has been done.

The last segment related to the spatial correlation of the ground motion predictions seems completely underdeveloped. If the authors intend to present it in a future article (as mentioned in L 348-349), then there is no point addressing this issue and drawing some concrete conclusions without any explanation within this article.

Reply to comment #3.

Section 4.2 "Spatial correlation structure of the predicted maps" has been moved in the Appendix. Lines 92-95 in the Introduction were modified as following: "Finally, evaluation of the spatial correlation structure was studied to provide the relation between local site effects and spatial resolution of ground motion maps; results of such analysis were not reported in the main text since it is out of the scope but preliminary results in terms of sill and range are reported in the Appendix referring to the seismic event occurred on October 30, 2016 (i.e., the strongest of the Central Italy seismic sequence).".

Therefore, this reviewer recommends a major improvement of the article before being considered for the publication. The main comments are as below:

**Scientific Comments:**

- The presentation of the datasets as maps would be helpful. Median and percentiles could be indicated on Figure 1 and 2 for a better comprehension.

Reply to comment #4.

A new Figure 1 with spatial distribution of the dataset was added to the manuscript.

Figs. 1 and 2 (Fig. 2 and 3 in the revised manuscript) were modified accordingly with median and percentiles of input and output data of the dataset.

[Figure]

Figure 1. Location of selected dataset (i.e., 1,435 accelerometric stations).

[Figure]

Figure 2. Distribution of input data for the training dataset.

[Figure]

Figure 3. Distribution of output data for the training dataset in terms of geoH IMs.

- • A) The method section must be improved. How are the outputs estimated from the input data. Which parameters have been used and how? B) Which proxy performed the best? What are the differences among different ML prediction models (Table 2) and why have they been adopted? C) How come the SD values are reduced by 45-60% (L217)? D) It's not clear how Openquake was used to determine the IMs – which site and fault parameters have been used for the calculation?

Reply to comment #5.

A) Please see Reply to comment #1.

B) Zhou et al. (2020) suggests that the parameters of the topographic elevation, the first gradient of the elevation and the second-order gradient in two orthogonal directions are enough to provide the acceptable topographic effect model.

So, the philosophy was to add these five proxies (topographic effect proxies) respect to the four standard proxies (magnitude, epicentral distance, hypocentral depth, $V_{s30}$) already used in the Ground Motion Models GMMs.

In the case of PGA, for example, adding these five proxies to the four standard proxies, performance improves (14% reduction in terms or Root mean squared error, RMSE, and 17% reduction in terms of Mean Absolute Error, MAE). **C)** In terms of residuals, there is a 46% reduction of standard deviation.

This table shows the details of the performance improvement for the PGA.

| Selected proxies | Root Mean Squared Error (RMSE) | Mean Absolute Error (MAE) | R-squared | mean of residuals | standard deviation of residuals |
|---|---|---|---|---|---|
| GMMs standard 4 proxies | 0.37 | 0.29 | 0.85 | -0.001 | 0.3 |
| This study 9 proxies | 0.31 | 0.24 | 0.89 | -0.000033 | 0.161 |

D) Official ShakeMap interpolates the values predicted by the GMM with the values of the stations, while this study results were compared with results (i.e., the external test) based on an equivalent method that uses only the forecast of the GMM. For this purpose, the Openquake software was used with these settings:

-calculation mode: scenario

- rupture_model_file = earthquake_rupture_model.xml (<magnitude>6.5</magnitude>

    <rake>-89</rake>  <hypocenter lat="42.8322" lon="13.1107" depth="9.2"/>

    <planarSurface strike="151" dip="47">

-intensity_measure_types = PGA, PGV, SA(0.3), SA(1.0), SA(3.0)

-truncation_level = 3.0

-gsim = BindiEtAl2011

- The reviewer disagrees with the statement in L 264-265. It's very difficult to visually determine if the maps in Figures 4 and 5 are coherent with geological and geomorphological characteristics.

Reply to comment #6.

Lines 264-265 were modified as "The map of Fig. 6 show an output that is in good agreement with the geophysical data (i.e., $V_{S30}$ in Fig. 7) and geomorphological data (i.e., h, $h_x$, $h_y$, $h_{xx}$, $h_{yy}$ not shown here for sake of brevity) and, therefore, highlights local site effects.".
The $V_{S30}$ map has been added to the manuscript (see Fig. 7).

[Figure]

Figure 7. $V_{S30}$ maps for the area of interest shown in Fig. 6. It can be noted that two extended areas in the southern part of the map (i.e., near Petrana and Torrita villages) are characterized by lowest values of $V_{S30}$ inducing the highest $Sa_{0.3}$ values (i.e., valley effect) as shown in Fig. 6.

- No such maps of the area have been presented. For comparison, the existing ground motion estimation for this area (e.g., from the shake map) and/or microzonation map could be presented on a same color scale.

Reply to comment #7.

ShakeMap (Fig. 8) and following lines were added to the text: "Fig. 8 shows the ShakeMap of $Sa_{0.3}$ (resolution 50 m x 50 m) regarding the Central Italy earthquake occurred on August 24, 2016 for the same area sketched in Fig. 7. As a general issue referring to the ShakeMap, the higher the distance from the epicentre (the star in Fig. 8) and the lower the predicted $Sa_{0.3}$. Hence, the ShakeMap does not provide ground motion variability induced by the local site condition (i.e., sub-soil setting and topography). In detail, ShakeMap provides $Sa_{0.3}$ equal to 0.36 g for the entire area of Arquata del Tronto (square A in Fig. 8) and equal to 0.99 and 1.08 g for Amatrice (square B in Fig. 8) as there is the data recorded by the AMT station.

On the other hand, the map proposed in this study provides ground motion variation also at the close-up scale (i.e., for urban area).".

[Figure]

Figure 8. ShakeMap (slightly modified from ShakeMap, 2021) of $Sa_{0.3}$ (resolution 50 m x 50 m) regarding the Central Italy earthquake occurred on August 24, 2016. A and B squares are referred to the close-ups at Arquata del Tronto and Amatrice, respectively. From the centre of the figure to the border, the homogenous coloured areas correspond to 1.20-1.50 g, 0.90-1.20 g, 0.60-0.90 g, 0.30-0.60 g, and 0.01-0.30 g intervals. It is evident that the map does not capture the variability at short distances.

- • The estimation of uncertainty presented in Figure 5 should be elaborated. What kinds of uncertainties have been considered and how have they been estimated?

Reply to comment #8.

The GPR model and the estimation of uncertainty of a GPR model is described here (https://it.mathworks.com/help/stats/gaussian-process-regression-models.html).

Because a GPR model is probabilistic, it is possible to compute the uncertainty using the trained model. GPR provides a probabilistic (Bayesian) approach for learning generic regression problems with kernels (Rasmussen and Williams, 2006). We used an exponential kernel function.

The kernel is thus parametrized by signal and noise hyperparameters. For training purposes, we assume that the observed variable is formed by noisy observations of the true underlying function. Moreover, we assume the noise to be additive independently identically Gaussian distributed with zero mean and variance. For prediction purposes, the GPR is obtained by computing the posterior distribution over the unknown output. Interestingly, this posterior can be shown to be a Gaussian distribution for which one can estimate the predictive mean (point-wise predictions) and the predictive variance (confidence intervals). The corresponding hyperparameters are typically selected by Maximum Likelihood, using the marginal likelihood (also called evidence) of the observations, which is also analytical. When the derivatives of the log-evidence are also analytical, which is often the case, conjugated gradient ascent is typically used for optimization (see Rasmussen and Williams, 2006 for further details).

A GPR model provides not only a prediction but also an uncertainty (or confidence) level for the prediction. Hence in contrary to other approaches (e.g. Neural Networks) uncertainty intervals are directly delivered along with mean estimates.

The uncertainty intervals estimated by GPR and through standard boostrapping of the nonlinear regression solution are compared in the figure below.

[Figure]

Note that GPR basically accounts for a reduction of uncertainty based on the relative local density of the input data points, not the outputs. This is obvious by looking at the predictive variance equation not here reported for sake of brevity. On the other hand, the variance of the bootstrap variance estimate quickly vanishes if one moves away from data points in the set. This behaviour correctly reflects the fact that the predictions will be practically zero far away from points in the training sets. Gaussian processes indicate that the uncertainty is high because no data has been observed in that area.

For these reasons in L 287-292 we comment in this way:

"It should be noted that the uncertainty is provided by a combination of the input data values. The uncertainty increases referring to input data values for which the ML is not well trained (Figs. 2 and 3 and discussion in § 2). For instance, std values around 0.3-0.4 are in the areas of inhabited villages, characterised by input data values widely represented in the training dataset, while values in the range 0.6-0.8 are observed in correspondence with the combination of high slope values and high $V_{S30}$ values, which are underrepresented in the training dataset." We have added a reference in the caption of fig. 9.

- The validation of the results seems insufficient. It uses the results from only one earthquake, for one spectral period and only 3 cross-sections. How are the Sa profiles (in Figure 6) obtained and modified from Giallini et al.? What about the ground motions at other spectral periods? Do they compare well? Other than these 3 cross-sections from two sites, how many more sites have been used to validate the results? How long are the cross-sections?

Reply to comment #9.

Work-flow of the adopted Machine Learning procedure discussed in Reply to comment #1 allows us to enlighten the difference between the validation phase and the comparison with different methodologies to predict ground motion. In detail, the validation phase was performed by means of k-5 fold cross validation. Testing phase was carried out referring to records of the October 30, 2016 Central Italy earthquake from 241 accelerometric stations. In addition, the comparison with different methodologies (ShakeMap, 2021, and 2D numerical simulations) allows us to show the ML capability to forecast the local scale ground motion variability despite ShakeMap (2021) and being in agreement with more advanced numerical simulations. Sa profiles by Giallini et al. and Gaudiosi et al. shown in this paper were provided by Giallini and Gaudiosi as personal communications.

- Segment 4.2 should be either eliminated or better explained. Results shown in Figure 7 and Table 5 should be interpreted in plain terms. The fact that the estimated short-scale variability is related to site conditions should be validated in terms of physical parameters. Which input parameters are causing these variations and to what extent? How close it is to the reality (example from a site)?

Reply to comment #10.

Section 4.2 "Spatial correlation structure of the predicted maps" has been moved in the Appendix. Following lines were reported in the Introduction of the paper: "Finally, evaluation of the spatial correlation structure was studied to provide the relation between local site effects and spatial resolution of ground motion maps; results of such analysis were not reported in the main text since is out of the scope but preliminary results in terms of sill and range are reported in the Appendix referring to the seismic event occurred on October 30, 2016 (i.e., the strongest of the Central Italy seismic sequence).".

[Figure]

**Consiglio Nazionale delle Ricerche (CNR)**
**Istituto di Geologia Ambientale e Geoingegneria (IGAG)**

**Editorial Comments:**

- The phrases 'referred to' or 'with reference to' have been heavily repeated and misused. All of them must be replaced by appropriate vocabulary or the sentences should be rewritten. The phrase 'bearing in mind' has been tediously repeated.

Reply:

The manuscript has been revised accordingly.

- The resolution of the map mentioned as '50 X 50 m' should be corrected as '50 m X 50 m' for the entire article.

Reply:

The resolution mention has been modified accordingly.

- For all the large numbers used in the article, the thousand separator should be corrected from 'XXX'000' to 'XXX,000' In English, comma separates the thousand.

Reply:

The thousand separator has been modified accordingly.

- Figure 2 has not been mentioned or explained anywhere. The cross-section labels on Figure 5 are not readable. The long web-address in the title could be replaced by a reference. The Figure 6 title should mention that the cross-sections are presented from Figure 5.

Reply:

The cross-section labels on Figure 5 (Fig. 9 in the revised manuscript) were modified (please see the figure reported here).

[Figure]

Figure 9. Ground motion prediction maps (Central Italy earthquake occurred on August 24, 2016) regarding the Arquata del Tronto (top) and Amatrice (bottom) in terms of $Sa_{0.3}$ mean value (left) and standard deviation (right) (resolution 50 m x 50 m). The base topographic layer was retrieved from Regione Marche (2021) and Regione Lazio (2021) for Arquata del Tronto and Amatrice maps, respectively.

The web-address in the caption has been replaced by a reference.

"Cross sections' locations are in Fig. 9." was included in the caption of Fig. 10 (Fig. 6 in the original form of the paper).

- L 108-112 should be rewritten. The current form is grammatically incoherent.

Reply:

For instance, focusing on Fig. 2 and M distribution, it results that the first and third quartile are 4.1 and 5.1, respectively. Moreover, focusing on elevation distribution, it results that the first and third quartile are 136 m and 761 m, respectively.

- In L 150 the phrase 'here adopted' should be corrected as 'adopted here'.

Reply:

The text has been modified accordingly.

- In L 232 'gauge the IMs' sounds unnatural. It could be replaced by 'estimate the IMs'.

Reply:

The text has been modified accordingly.

- The webpage links should be referred to as a reference with last access date. It is sufficient to cite them as references in the text rather than citing the link everywhere.

Reply:

The webpage links have been referred to as a reference with last access date.

L 376 and 385 are not expressed as full sentences. Either they should be turned into paragraph subtitles by using a colon (:) after 'following' in L375 or they should be rewritten as full sentences. Similar remark for L 397-403.

Reply:

The text has been modified accordingly.

---

## Author Response (AR2)

Title: "Ground motion prediction maps using seismic microzonation data and machine learning" by Federico Mori et al., Nat. Hazards Earth Syst. Sci. Discuss.,

**Editor comment**

Dear authors,

this manuscript can now be accepted - but, please, implement still some minor corrections suggested by the reviewers before we provide the version to be processed for in press.

Sincerely yours

Hans-Balder Havenith, assoc. editor

**Dear Editor,**

we thank you and reviewers for their thorough assessment of our paper. We have carefully addressed the reviewers' comments and made corresponding changes to the manuscript. We have modified and added some figures. We have carefully revised the ambiguous statement in the article. The response to reviewers' comments has been reported below.

**Referee #1 comment**

The authors would like to thank the reviewer for his comments.

The manuscript has been improved. This paper involves lots of work, it would be a plus to provide some insights. One point mentioned by both reviewers is about the strength of correlation between each input and output, answering this question will give us some knowledge of the complex pattern within the data to render the ML models somewhat interpretable. Could the authors provide a plot on the feature importance from which we could see which predictor variable is necessary or unnecessary in the model. This is also to validate the findings of Zhou et al (2020) whether the pattern reported in China on topographic effect modeling is still applicable in Italy.

**Reply:**

We have inserted a sentence at the end of Section 3.1 and referred back to the appendix for the plot.

**Referee #3 comment**

The authors would like to thank the reviewer for his comments.

The paper shows the approach to derive seismic parameters through ML approach. I read the manuscript and checked the modifications carried out following past reviewers comments and I found the Authors' effort satisfactory.

**Referee #4 comment**

The authors would like to thank the reviewer for his comments.

This paper aims to assess the advantages / limitations of machine learning techniques - with respect to more conventional approaches ie. GMM - to produce ground motion prediction maps for large areas considering both topographic and stratigraphic local conditions. To do so, the authors identified key-parameters on which they develop their methodologies. In addition, they provide a comparison of their ground motion prediction maps to the results of advanced numerical simulations based on detailed sub-soil models.

This reviewer acknowledges that the authors took into consideration some of the guidelines of the reviewers of the first step of the review process, which is good. Answering the remarks / questions below might help to further improve the quality of this paper which, in the present state, still requires minor revisions according to this reviewer.

Scientific comments:

Introduction:

- Line 26: « regarding the spatial correlation ... » the meaning of the spatial correlation in this context is not clear to this reviewer

**Reply:**

The following statement was added: "where the spatial correlation is the spatial characteristics of the ground motion arising from similarities in the seismic wave paths and local-site effects".

- Line 81: as already pointed out by reviewers in the first step of the review process, this paragraph should be erased because it is not commented in the text and it does not help to better understand the work.

Reply:

These lines were moved to the section "Discussion and conclusions" since they are briefly presented in the appendix and are outlined as future research.

- Line 85: this paragraph could be part of the discussion

**Reply:**

These lines were moved to the section "Discussion and conclusions".

**Data:**

- Line 96: « some data distributions seem to be imbalanced »: the meaning of this sentence is not clear to this reviewer

**Reply:**

As already written in line 94: An imbalanced training input dataset is characterised by an unequal distribution of values.

**Method:**

- Line 61: is Vs30 parameter enough to represent sub-soil conditions and take into account possible geological site effects

**Reply:**

"VS30, the fundamental frequency of the deposit ( $f_0$ ), and the depth to the engineering bedrock (H800) are the key-parameters which well gauge the effect of local sub-soil conditions on the seismic wave propagation (i.e., lithostratigraphic effect)" was already reported in the manuscript. The following statement was added consistent to what already reported in "Introduction": "The only VS30 was used in the adopted ML approach since the Italian VS30 map was provided by Mori et al. 2020a while national  $f_0$  and H800 maps are not currently available".

- Line 194: the references should be moved to Line 184 according to this reviewer.

**Reply:**

Lines were moved accordingly.

**Results :**

- Line 277: this statement of a good agreement between geophysical data and ground motion prediction maps is not convincing to this reviewer. In addition, it would be worth showing also the geomorphological maps to allow the authors to draw such conclusions. As previously mentionned by

reviewers in the first step of the review process, it is very difficult to visually determine if both maps are coherent (see Accumoli or Amatrice for instance and compare their values to Petrana). Maybe a different definition of the colorbar scales in both Figures could ease the comparison between plots.

**Reply:**

The required maps have been added to the appendix.

- Line 295 : according to this reviewer, the comparison with numerical modeling results, although worthy of investigation, is not described into sufficient details to allow the readers to compare ML results to more advanced numerical simulations results. Some criticisms were already made by reviewers during the first step of the review process. Because this section of the comparison could be an original / novel contribution of the paper, to this reviewer it deserves further explanations : methodology used in the numerical modeling, materials behaviour and properties, mesh resolution, seismic inputs parameters, etc.

Besides, the correspondance between numerical results and this study should be quantified to this reviewer: looking at the shape of the curves, the fit is not so obvious. Although 0.3s might be of interest for this area, what about the response of both methods at other spectral periods ?

**Reply:**

We apologize since further explanation of the numerical modelling cannot be reported in the manuscript since they have been performed by different research groups not including all the authors of this manuscript. The research papers were already cited in the manuscript Gaudiosi et al. (2021), Giallini et al. (2020), and Grelle et al. (2020). The response of numerical method at other spectral periods was not included in the manuscript since not available to us (i.e., not provided to us by the authors of the mentioned works).

Correspondence between numerical results and this study was considered valuable in terms of trend rather than by a quantitative point of view as already stated in lines "It should be noted that the trend of the values of our study reproduces that of the numerical simulations". We would emphasize that the qualitatively good results of our work are the peak values of Sa trends due to the presence of a crest of the topography and of a buried valley as in the case of Amatrice (section AA' in Figure 10) and Arquata del Tronto (section CC' in Figure 10), respectively. On the contrary, the ShakeMap cannot reproduce the ground motion variation induced by the local variation of site conditions.

Finally, the following table and statement was added to the manuscript since the real records of OSS site and AMT station were considered the benchmark for the validation of our results. Future validation will be considered since we are selecting denser station array than the currently available.

"The difference between different methodology and the recorded values were quantified according to the following Equation and are provided in **Table 1**.

 $\epsilon_{_{Sa}} = \frac{Sa_{_{0.3 \text{ estimated}}} - Sa_{_{0.3 \text{ recorded}}}}{Sa_{_{0.3 \text{ recorded}}}} \cdot 100$

**Table 1. Difference $\epsilon$ in percentage between Sa0.3 determined by means of different methodologies and recorded values for Central Italy earthquake occurred on August 24, 2016.**

|            | Section BB'     | Section CC'     |
|------------|-----------------|-----------------|
|            | ε Sa | ε Sa |
|            | (%)             | (%)             |
| this study | -7              | -14             |
| Numerical  | -43             | -44             |
| ShakeMap   | -42             | -40             |

**Discussion / conclusion**

- Line 371 : « it is possible to predict the losses in the area struck by the earthquake in near real time » : the aim of this paper was to produce ground-motion prediction maps over large areas based on ML technique. As mentioned in the text, the link between ground motions and damage is not straightforward. Therefore the content of this sentence calls for some nuance.

**Reply:**

We have deleted damage and inserted "shaking scenarios".

Technical corrections :

- Line 11: what is a « burried morphology »? Example ?

**Reply:**

"(e.g., irregular sub-interface between soft and stiff soils)" was added to the text.

- Line 56: « capable of properly representing ... »

**Reply:**

Modified accordingly.

- Line 69: « in terms of »

**Reply:**

Modified accordingly.

- Line 104 « it seems a hard task to improve the training dataset » ; line 105 « anyway » : to this reviewer, this is more spoken language

Reply:

Line was modified as following: "Hence, the training dataset cannot actually be improved.".

"Anyway" was deleted.

Figures / tables:

- Table 1: hxx refers to the « second order partial derivative of dxx » ? Same for hyy ? Please check.

Reply:

Modified accordingly.

- Figures 2 and 3: replace 'o' by 'th' in the percentiles definitions (25 and 75). Add units of the parameters (25 th, median and 75 th) when relevant.

Reply:

"" was replaced by "th". The units were not added to (25 th, median and 75 th) since it is already reported on the x-axis title.

- Figure 4: erase « performance performance »

Reply:

Modified accordingly.